# Liquid condensation of reprogramming factor KLF4 with DNA provides a mechanism for chromatin organization

Rajesh Sharma[1,8], Kyoung-Jae Choi[1,8], My Diem Quan[1], Sonum Sharma[1], Banumathi Sankaran [2], Hyekyung Park[4], Anel LaGrone[4], Jean J. Kim [3,4], Kevin R. MacKenzie[1,5,6 ✉], Allan Chris M. Ferreon [1 ✉], Choel Kim [1,7 ✉] & Josephine C. Ferreon [1 ✉]

Expression of a few master transcription factors can reprogram the epigenetic landscape and three-dimensional chromatin topology of differentiated cells and achieve pluripotency. During reprogramming, thousands of long-range chromatin contacts are altered, and changes in promoter association with enhancers dramatically influence transcription. Molecular participants at these sites have been identified, but how this re-organization might be orchestrated is not known. Biomolecular condensation is implicated in subcellular organization, including the recruitment of RNA polymerase in transcriptional activation. Here, we show that reprogramming factor KLF4 undergoes biomolecular condensation even in the absence of its intrinsically disordered region. Liquid–liquid condensation of the isolated KLF4 DNA binding domain with a DNA fragment from the *NANOG* proximal promoter is enhanced by CpG methylation of a KLF4 cognate binding site. We propose KLF4-mediated condensation as one mechanism for selectively organizing and re-organizing the genome based on the local sequence and epigenetic state.

[1] Department of Pharmacology and Chemical Biology, Baylor College of Medicine, Houston, TX, USA. [2] Molecular Biophysics and Integrated Bioimaging, Berkeley Center for Structural Biology, Lawrence Berkeley National Laboratory, Berkeley, CA 94720, USA. [3] Department of Molecular and Cellular Biology, Baylor College of Medicine, Houston, TX, USA. [4] Center for Stem Cells and Regenerative Medicine, Baylor College of Medicine, Houston, TX, USA. [5] Department of Pathology and Immunology, Baylor College of Medicine, Houston, TX, USA. [6] Center for Drug Discovery, Baylor College of Medicine, Houston, TX, USA. [7] Verna and Marrs McLean Department of Biochemistry and Molecular Biology, Baylor College of Medicine, Houston, TX, USA. [8] These authors contributed equally: Rajesh Sharma, Kyoung-Jae Choi. ✉email: km5@bcm.edu; allan.ferreon@bcm.edu; ckim@bcm.edu; josephine.ferreon@bcm.edu

Krüppel like factor 4 (KLF4) is a key constituent of reprogramming cocktails that transform fibroblasts to induced pluripotent stem cells (iPSCs)[1–4]. KLF4 cooperates with transcription factors (TFs) OCT4 and SOX2 in reprogramming to silence somatic enhancers and activate enhancers of pluripotency genes[5,6], including the 'gateway to pluripotency' gene *NANOG*[7], which is highly expressed in embryonic stem cell (ESCs)[8–11]. In PSCs, KLF4 is enriched at the *NANOG*[12] and *OCT4*[13] loci, which interact through space with many other pluripotency-related genomic sites. KLF4 is enriched at ESC super-enhancers[14] and at iPSC genomic anchors that make more than four contacts, further implicating KLF4 in chromatin organization[15]. How KLF4 or other TFs might initiate chromatin reorganizations that determine cell fate is of intense interest[3,15,16].

KLF4 contacts the DNA major groove with three tandem $C_2H_2$ zinc fingers (ZnFs) that make specific interactions[17,18] at 9 base pair (bp) cognate DNA sites[19,20]. The first 400 KLF4 residues are likely to be disordered because they have low sequence complexity, and intrinsically disordered regions (IDRs) of other TFs help to silence[21,22] or activate[23–26] gene expression. In current models for transcriptional activation, TFs bound to their cognate sites cooperate with co-localized co-activators to recruit Mediator complex and RNA polymerase II through IDR:IDR mediated biomolecular condensations[23–26]. The KLF4 DNA binding domain and IDR might participate in such processes in open chromatin, and the KLF4 preference for CpG methylated over unmethylated cognate sites[27] combined with its ability to bind 6 bp partial sites in nucleosomal DNA[16] could help target it to silenced chromatin. The ability of KLF4 to undergo biomolecular condensation could facilitate pioneer interactions with closed chromatin and, as others have speculated[15], might stabilize long-range contacts between genomic loci.

Here, we show that KLF4 forms a liquid-like biomolecular condensate with DNA that recruits OCT4 and SOX2. Surprisingly, the intrinsically disordered region is not essential for KLF4 condensation in cells, and a KLF4 fragment comprising the isolated DNA binding domain (DBD) condenses with DNA in vitro. KLF4 DBD condensation with a *NANOG* promoter duplex is strongly enhanced by CpG methylation of a KLF4 cognate site, and ZnF point mutations that weaken interactions with DNA cognate sites decrease condensation in cells and in vitro. Single molecule methods show that KLF4 tandem zinc fingers bring together short DNA duplexes in dilute solution by a bridging interaction. We propose that bridging and/or condensation with DNA in a sequence- and CpG methylation-dependent manner underlie KLF4 function as a key chromatin organizer and pioneer transcription factor in somatic cell reprogramming.

## Results

**KLF4 forms nuclear condensates at modest expression levels.** We used expression tags to monitor the distribution of KLF4 by fluorescence microscopy in HEK 293T cells or BJ fibroblasts, the somatic cells most widely used for reprogramming[28]. KLF4 fused to mTurquoise2 (KLF4-mTurq) localizes to the nucleus and forms small puncta or round droplets, whereas mTurquoise2 alone is diffusely distributed throughout the nucleus and the cytoplasm (Fig. 1a). Transfection produces cells with various expression levels of tagged protein; KLF4-mTurq distribution is diffuse at the lowest expression levels, but most cells that express detectable KLF4-mTurq show punctate expression or droplets (Fig. 1b). Because round droplets are hallmarks of liquid–liquid phase separation (LLPS)[29,30], we monitored KLF4-mTurq fluorescence after photobleaching; fluorescence recovers rapidly in both large droplets and small puncta in BJ fibroblasts (Fig. 1c), indicating that KLF4-mTurq in the condensate diffuses rapidly

and is therefore liquid-like. Time courses using 3D z-stack fluorescence imaging reveal fusion of small droplets in BJ fibroblasts (Fig. 1d), indicating a liquid-like KLF4-mTurq condensate. Treatment with 1,6-hexanediol largely dissolves the KLF4-mTurq puncta and round droplets in HEK 293T cells (Fig. 1e), consistent with a liquid-like condensate[31].

A KLF4-mCherry fusion expresses at lower average levels than KLF4-mTurq but also forms puncta and droplets (Supplementary Fig. 1), indicating that the identity of the expression tag is not critical to condensation. Endogenous KLF4 levels have not been reported, but the KLF4-mTurq levels quantified by brightness (0.7 μM average for cells with puncta; 2.5 or 4.0 μM average for cells with small or large droplets, respectively; see Supplementary Fig. 2) are similar to those reported for TFs SOX2 or OCT4[32]. We expect that KLF4 expression driven by vectors in reprogramming cocktails[33] would result in robust biomolecular condensation.

**The intrinsically disordered region is dispensable for KLF4 condensation.** To identify domains that contribute to KLF4-mTurq biomolecular condensation, we expressed constructs lacking either the IDR (residues 1–417) or the DNA binding domain (DBD; residues 418–513) (Fig. 2a). KLF4$^{\Delta DBD}$-mTurq, which lacks the three tandem ZnFs, expresses well, is diffusely distributed throughout the cytoplasm and nucleus, and only rarely forms nuclear puncta (Fig. 2b, top). KLF4$^{\Delta IDR}$-mTurq, which lacks the low complexity region, expresses poorly, localizes to the nucleus, and forms droplets similar to KLF4-mTurq (Fig. 2b, bottom). Scoring cells for the presence of puncta and plotting them by total mTurq brightness reveals diffuse distribution at the lowest expression levels; KLF4-mTurq and KLF4$^{\Delta IDR}$-mTurq mutant form puncta at similar modest expression levels, whereas KLF4$^{\Delta DBD}$-mTurq forms puncta only at high expression levels (Fig. 2c). The dispensability of the IDR indicates that the DBD alone can drive KLF4 condensation, but the tag, which comprises most of KLF4$^{\Delta IDR}$-mTurq (Fig. 2a), may contribute in some way. To test directly whether the KLF4 tandem zinc fingers drive biomolecular condensation, we studied the isolated domain in vitro after expression in *E. coli*.

**The KLF4 DNA binding domain phase separates with cognate DNA.** Purified KLF4 DBD is readily soluble and does not condense or precipitate at physiological salt or upon addition of 10% PEG 8000, a crowding agent used to enhance the weak interactions that drive biomolecular condensation (Fig. 2d). Because proteins that bind RNA can undergo RNA-induced phase separation[34], we tested the ability of NANK, a 30 bp *NANOG* promoter DNA duplex containing 3 KLF4 cognate sites, to induce phase separation of DBD. Adding 1 μM NANK to 6 μM DBD in physiological salt, without PEG, results in droplets that are visible by bright field microscopy (Fig. 2d, right). This DNA-induced condensation occurs without any labels or tags on either the isolated DBD or the DNA duplex. To determine if DBD and NANK co-localize, we labeled DBD with Alexa Fluor 594 (AF594) and mixed it with NANK in the presence of the dye YOYO-1, which binds DNA with high affinity[35]. Two-channel fluorescence images confirm that DBD-AF594 and NANK-YOYO-1 co-localize in all droplets (Fig. 2e). Time courses of NANK-induced DBD condensation shows that droplets form, grow, fuse, settle, and wet the bottom surface (Fig. 2f), indicating that the phase is liquid-like. The liquid nature of large droplets is confirmed by rapid recovery of fluorescence after photobleaching localized regions (Fig. 2g). We conclude that DBD undergoes liquid–liquid phase separation (LLPS) with NANK at physiological salt without the need for crowding agents.

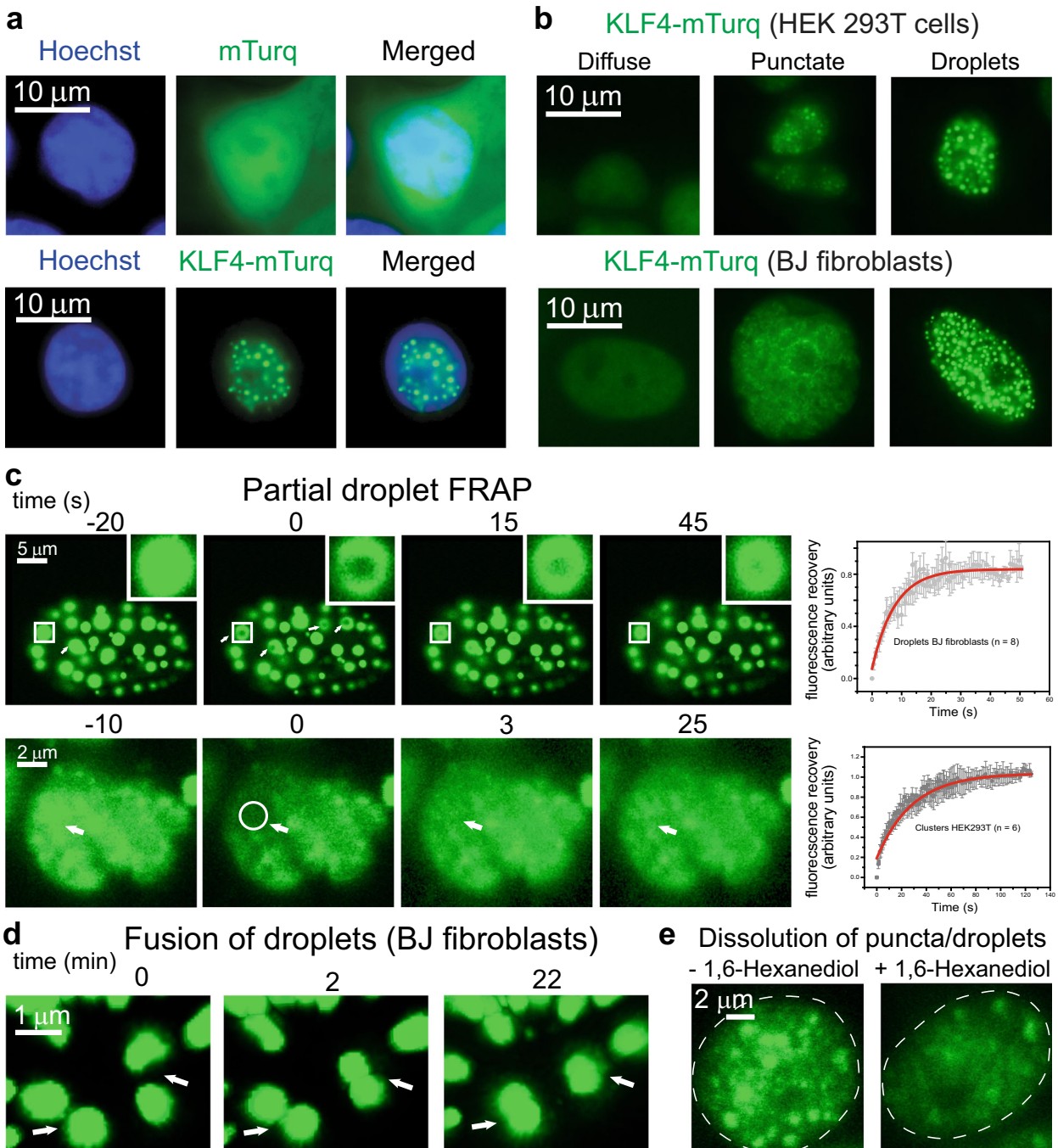

**Fig. 1 KLF4 forms a condensed liquid phase in HEK 293T cells and BJ fibroblasts. a** Fluorescence microscopy images of DNA (Hoechst), mTurq fluorescent tag (top), and KLF4-mTurq (bottom) in HEK 293T cells. Nuclear mTurq distribution is diffuse whereas KLF4-mTurq forms biomolecular condensates. Similar results were obtained for >5 biological replicates. **b** HEK 293T cells (top) and BJ fibroblasts (bottom) expressing KLF4-mTurq exhibit diffuse distribution (left), irregular puncta (middle), or droplets (right; circularity > 0.8). Similar results were obtained for 3 biological replicates. **c** Fluorescence recovery after photobleaching (FRAP) of KLF4-mTurq droplets in BJ fibroblasts (top row) bleached at positions indicated by white arrows; right panel is a recovery curve ($n = 8$ droplets) and enlarged insets track one droplet (white square). FRAP of KLF4-mTurq puncta in HEK 293T cells (bottom row) bleached at positions indicated by white arrows and circle; right panel is a recovery curve ($n = 6$ punctate fields). Data are presented as mean values ± SD. **d** Fluorescence image time course of droplet fusion (at white arrows) in BJ fibroblasts. Fusion was verified with 3D z-stack images. **e** Fluorescence microscopy image of KLF4-mTurq puncta and droplets in HEK 293T cells before (left) and after (right) 1,6-hexanediol treatment. Nucleus outline in white dashes. Similar results were obtained for 2 biological replicates.

DBD:NANK LLPS depends in complex ways on the component concentrations. At 6 μM DBD, 0.25 μM NANK induces readily detectable LLPS, and increasing NANK up to 1 μM increases the amount of condensate, but further increases in NANK actually produce less condensate, and at 3 μM NANK and above LLPS is no longer detected (Fig. 2h). The lack of LLPS at high NANK suggests that phase separation requires a minimum DBD:NANK ratio, perhaps to saturate NANK cognate KLF4 sites. At any given DBD concentration, LLPS is not observed below a threshold NANK concentration; this NANK threshold is lower at

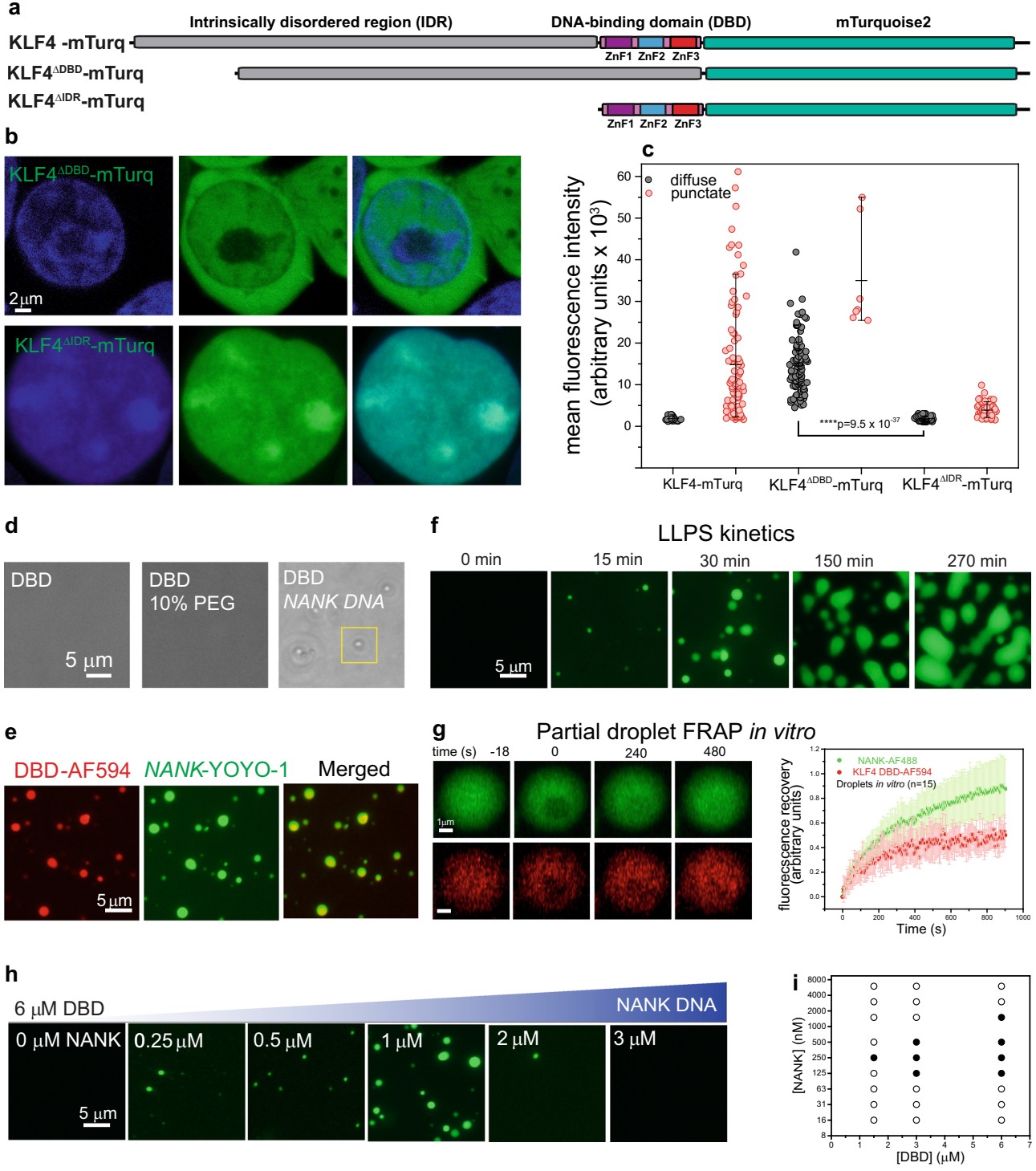

high DBD levels (Fig. 2i, Supplementary Fig. 3). These data describe a phase diagram for DBD:NANK LLPS (Fig. 2i) and challenge us to understand the nature of DBD:NANK interaction.

**KLF4 DBD forms a 3:1 complex with a *NANOG* promoter duplex.** KLF4 binds 9 bp cognate sites containing methylated GGCG or 'intrinsically methylated' GGTG[17] and activates expression from the *NANOG* promoter through two GGTG elements[36] (sites KLFA and KLFC in Fig. 3a). A GGCG element in the *NANOG* promoter reverse strand (KLFB in Fig. 3a) is

methylated and silenced in germ cells[37], and *NANOG* promoter hypermethylation must be reversed to achieve pluripotency[38]. Because the nature of KLF4 DBD binding to this DNA might be important for LLPS, we tested binding at these sites in vitro. Each of three 12 bp duplexes excerpted from the 30 bp *NANOG* promoter fragment NANK contains a central GG(C/T)G and binds DBD in electromobility shifts assays (EMSA) (Supplementary Fig. 4). EMSA titration of the 30 bp fragment NANK (or its CpG methylated variant, NANKm) with DBD gives complexes of three different mobilities, consistent with DBD forming 1:1, 2:1, and 3:1 complexes with these DNAs (Fig. 3a). At the same DNA

**Fig. 2 The KLF4 intrinsically disordered region is dispensable for biomolecular condensation. a** Domain organization of KLF4-mTurq constructs. **b** Fluorescence microscopy of mTurq fusions and DNA (Hoechst) in HEK 293T cells. KLF4$^{\Delta DBD}$-mTurq (top) is found throughout cells and is diffuse in the nucleus; KLF4$^{\Delta IDR}$-mTurq (bottom) localizes to the nucleus and forms condensates. Similar results were obtained results for 2 biological replicates. **c** HEK 293T cells expressing KLF4-mTurq variants classified as diffuse (gray) or punctate (red) are plotted by their mean fluorescence intensity; the long horizontal bars mark the median, and the top and bottom bars on the vertical lines denote the 90-10 percentiles. Diffuse cells expressing deletion constructs show very different fluorescence levels (statistical tests were performed using a two-sided Student's paired $t$-test; $P$ value shown; $n = 105$ cells each). KLF4$^{\Delta IDR}$-mTurq expresses poorly and 48% of cells are punctate ($n = 97$); KLF4$^{\Delta DBD}$-mTurq expresses well and 6% of cells are punctate ($n = 7$). Similar results were obtained for 2 biological replicates. **d** Bright field microscopy of 6 μM KLF4 DBD (left), with 10% PEG-8K (center) or 1 μM NANK (right) in TS buffer (12.5 mM Tris, 70 mM NaCl, pH 7.4). **e** Fluorescence microscopy of 6 μM DBD in TS buffer (with 50 nM DBD-AF594; red) and 1 μM NANK (with 100 nM YOYO-1; green); yellow droplets in merge indicate colocalization. Similar results were obtained for 2 replicates. **f** Fluorescence microscopy of DBD:NANK droplet time course taken at a focal plane close to the surface. Droplets form, grow, fuse and wet the surface. Similar results were obtained for >10 replicates. **g** FRAP of droplets monitoring NANK-AF488 (green; top row) or KLF4 DBD-AF594 (red; bottom row). Curves (at right) show mean and standard deviation for 15 droplets. **h** DNA concentration dependence of DBD:DNA liquid–liquid phase separation (LLPS). Fluorescence microscopy images of 6 μM DBD with NANK DNA (0–3 μM) in TS buffer with 100 nM YOYO-1 (green) after 30 min incubation. **i** LLPS measurements at various DBD and NANK concentrations; solid/open circles indicate LLPS/no LLPS. Images are scored as LLPS if the coefficient of variation (CV, standard deviation/mean pixel intensity) is >0.2 and the mean fluorescent intensity is >0.4 arbitrary units. Values were determined using ImageJ for 2–3 independent replicates.

## Table 1 Data and refinement statistics.

|  | KLF4:NKA |
|---|---|
| *Data collection* |  |
| Space group | $P\ 1\ 2_1\ 1$ |
| Cell dimensions |  |
| $a, b, c$ (Å) | 38.96, 46.69, 45.20 |
| $\alpha, \beta, \gamma$ (°) | 90.0, 113.5, 90.0 |
| Resolution (Å) | 22.17–2.14 (2.22–2.14)[a] |
| $I/\sigma(I)$ | 4.31 (1.23)[a] |
| Completeness (%) | 96.14 (94.20)[a] |
| Redundancy | 3.9 (3.8)[a] |
| $R_{means}$ | 0.2289 (2.774)[a] |
| $CC_{1/2}$ | 0.975 (0.304)[a] |
| *Refinement* |  |
| Resolution (Å) | 22.17–2.14 (2.22–2.14)[a] |
| No. reflections | 8016 (780)[a] |
| $R_{work}/R_{free}$[b] | 0.1975/0.2474 |
| No. atoms |  |
| Proteins | 1142 |
| Ligand/ion | 5 |
| Water | 93 |
| B-factors |  |
| Protein | 48.64 |
| Ligand/ion | 47.56 |
| Water | 45.46 |
| R.m.s. deviations |  |
| Bond lengths (Å) | 0.009 |
| Bond angles (°) | 1.03 |

[a]Values in parentheses are for the highest resolution shell.
[b]5.0% of the observed reflections were excluded from refinement for cross validation purposes.

concentrations, 3 DBD equivalents form a detectable 3:1 complex with NANKm, but 6 equivalents are needed to form such a complex with NANK, consistent with the KLF4 preference for GG$^m$CG over GGCG[27]. We conclude that our KLF4 DBD preparations are well folded and show target selectivity, and that DBD can form a 3:1 complex with NANK.

**KLF4 DBD forms a 1:1 complex with a cognate *NANOG* dodecamer.** We determined the crystal structure of the 1:1 complex of DBD bound to a dodecamer containing the *NANOG* proximal promoter KLFA site (Fig. 3a, Table 1). As with previous KLF4 crystal structures with DNA[17,18], each ZnF contacts three or four base pairs in the DNA major groove (Fig. 3b). The overall structure of the DBD:NKA complex is similar to previous

KLF4:decamer complexes[18] and each ZnF makes base-specific contacts mediated by one or more 'specificity residues' at positions −1, 2, 3, and 6 of the canonical C2H2 recognition code[39]. Residues R473 and R479 of ZnF2 (in positions −1 and 6) and R501 of ZnF3 (in position −1) hydrogen bond with the N7s and O6s of bases G5, G6, and G8, respectively; the arginine side chains also make polar interactions with water, ions, and/or aspartate side chains (Fig. 3c). As in previously solved structures, ZnF1 makes fewer base-specific contacts than ZnF2 or ZnF3. The only base-specific contact made by ZnF1 in this structure is H446 (in position 3) hydrogen bonding to N2 of base G10 (Fig. 3c). With other target DNAs, K443 (position −1) of ZnF1 contacts a G[17,18]; our target has a C at the corresponding position 11 in our dodecamer, and the K443 side chain is disordered in our structure. Interactions between a glutamate (E476, position 3) and the methyl group of T7 (Fig. 3c) in NKA are similar to those seen for KLF4 DBD bound to a DNA decamer containing methylated-C (PDB ID: 4M9E). KLF4 DBD binds to the GGTG of our dodecamer with the same conformation as it does to the GG$^m$CG of a decamer (87 Cα-atoms superimpose with a root-mean-square deviation of 0.97 Å). When bound to a DNA heptamer (PDB ID: 2WBS), the ZnF2 and ZnF3 domains contact their target bases as in the decamer (354 atoms superimpose with a root-mean-square deviation of 0.48 Å), but ZnF1 adopts a new orientation[18] (Fig. 3d) that has been invoked to explain a 6 bp consensus KLF4 binding site on nucleosomes[16].

**KLF4 site overlap may drive non-canonical binding.** The three NANK KLF4 sites match the 9 base KLF4 consensus site from JASPAR[40] at 6 or 8 positions, but because the KLFA and KLFB sites overlap (Fig. 4a), how a 3:1 DBD:NANK complex forms (Fig. 3a, right) is not clear. We sought to determine the structure of three DBDs bound to NANK (or NANKm), but preparing 3:1 complexes invariably gave condensation, not crystallization. We, therefore, used superpositioning to determine if the canonical interactions seen in the DBD:NKA complex can be accommodated at sites KLFA, KLFB, and KLFC in a B-DNA model of the *NANOG* proximal promoter. Individual superpositions at each site suggest favorable protein:DNA contacts, and simultaneously placing DBD at KLFC and either KLFA or KLFB generates no clashes (Fig. 4b). However, canonical DBD occupation of both KLFA and KLFB causes intermonomer clashes of the two modeled ZnF1 domains (Fig. 4c). We conclude that in the observed 3:1 complexes (Fig. 3a), at least one ZnF1 projects away from the DNA. Using the pose for DBD bound to a DNA heptamer (Fig. 3d, black) at KLFA or KLFB with a canonical DBD posed at the other site relieves the clash (Fig. 4c). The four residue ZnF1–ZnF2

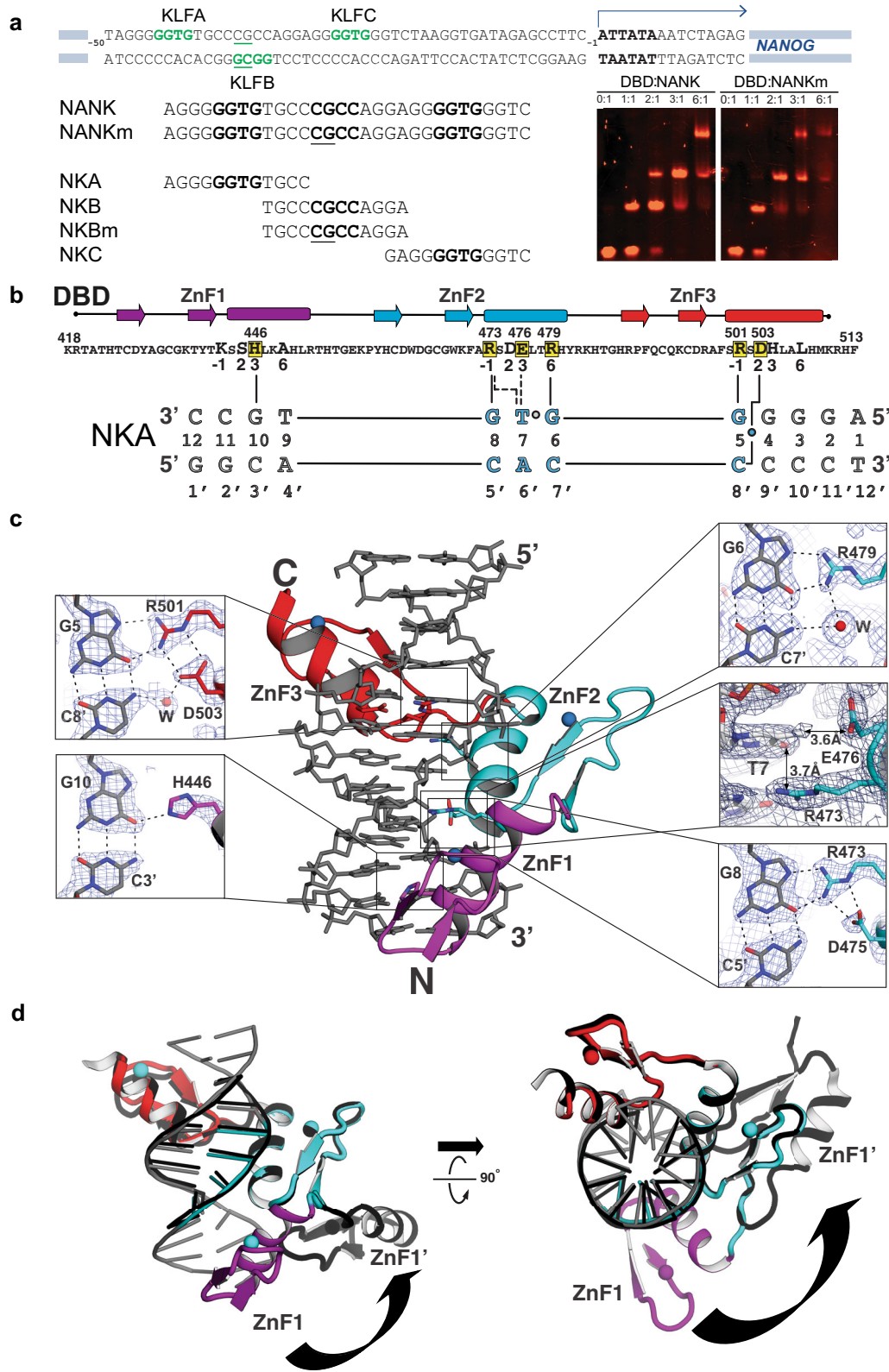

linker should allow ZnF1 to adopt different orientations, consistent with tandem ZnFs acting as independently folded "beads on a string"[41]. KLF4 would contact NANK with ZnF2 and ZnF3, using the 6 bp KLF4 binding site detected on nucleosomal DNA[16] and retaining most of the basis for specificity inferred from structural analysis[18].

If overlapping KLF4 sites have functional importance because they obligately expose a ZnF, then such arrangements should be conserved. The mouse *NANOG* promoter (Supplementary Fig. 5) contains four overlapping KLF4 cognate sites in the −90 to −65 region: two sites on the top strand contain GGTG (and match the 9 bp consensus at 7 or 8 positions), and one site on each strand

**Fig. 3 Structural determinants of DBD:NANK interactions. a** Schematic of the *NANOG* promoter (top) and studied DNA duplexes (bottom left) with GG(T/C)G sites in bold and $^{m}$CpG sites underlined. Electrophoretic mobility shift assays (bottom right) with 0–6 µM DBD and 1 µM DNA (NANK or NANKm) were stained with EtBr (red) for two independent replicates. **b** DBD contacts with the NKA dodecamer. (Top) DBD sequence and secondary structure for ZnF1 (magenta), ZnF2 (cyan) and ZnF3 (red). (Bottom) NKA sequence numbered 5′-to-3′; GGTG motif in cyan. DBD residues that contact DNA bases are highlighted in yellow; numerals −1, 2, 3, or 6 refer to the canonical C2H2 recognition code[39]. Solid (or dotted) vertical lines indicate hydrogen bonds (or van der Waals contacts). **c** Crystal structure of DBD bound to the NKA dodecamer (middle) and close-up views (sides) with $2F_{o}$–$F_{c}$ electron density maps contoured at 1.0 σ. Three ZnFs, colored as in (**b**), wrap around the DNA (gray) with zinc ions as blue spheres, residues and bases colored by atom type (carbon, black; nitrogen, blue; oxygen, red), hydrogen bonds as black dotted lines, and van der Waals contact distances marked with black arrows. **d** Superposition of DBD:NKA complex (same colors as above) and a previous DBD:DNA complex (PDB ID: 2wbs, black) shown in two views related by a 90° rotation. ZnF1 in 2wbs is rotated away from the DNA axis compared to our complex (arrow).

contains GGCG (and matches the consensus at 6 or 8 positions). DBD binding schematics for these sites suggest that the mouse *NANOG* promoter would also exhibit a tethered, continuously solvent-exposed ZnF1 when saturated with KLF4, and that binding could be modulated by CpG methylation. Since tight DNA binding can be achieved by two ZnFs[42] or one ZnF flanked by a basic region[43], we hypothesized that the exposed ZnF1 might recruit a second DNA partner, and that such DNA bridging might drive biomolecular condensation.

**KLF4 DBD can bridge two DNA duplexes**. We tested this hypothesis using single molecule Förster resonance energy transfer (smFRET)[44–46]. In DBD:NANK models where ZnF1 is excluded from either KLFA or KLFB (Fig. 4d), the 5′ end of the NANK coding strand is 40–43 Å from the Cα of H446, a ZnF1 residue that canonically contacts the major groove. We therefore 5′ end labeled the NANK coding strand with fluorescent donors or acceptors, reasoning that close proximity of the excluded ZnF1 might enable it to bring another labeled DNA close enough for FRET (Fig. 4e). With two labeled DNAs co-dissolved at low concentrations (100 pM AF488-labeled NANK, 500 pM AF594-labeled NANK), donor emission is observed but FRET is not (Fig. 4f, left). FRET events induced by 1 µM unlabeled KLF4 DBD (Fig. 4f, right) show that DBD can bring two DNAs together (closer than 55 Å, the $R_0$ for this label pair), providing a mechanism for DBD biomolecular condensation with DNA. Although detected at dilute concentrations that do not support mesoscale LLPS (Fig. 2i), these events are likely those that drive condensation at higher concentrations. The smFRET data do not define the stoichiometry of the complex(es), but they are consistent with our non-canonical model for DBD:NANK interaction (Fig. 4e).

**Non-cognate DNA can drive KLF4 DBD phase separation**. If bridging between NANK molecules by a single continuously excluded ZnF can drive LLPS, then DBD bound to non-cognate sites might make similar bridging interactions when one ZnF transiently leaves the major groove. To test this, we mixed DBD with four DNA duplexes of 12 to 40 bp that lack a GG(C/T)G. None induce LLPS at 3.0 µM DBD and 0.25 µM DNA, conditions at which *NANK* readily drives LLPS, but at 10 µM DBD and 3 µM duplex, all four non-cognate DNAs produce phase separated droplets (Fig. 5a). We infer that DBD bound to non-cognate DNA samples binding modes that transiently expose ZnFs to interact with another duplex.

DBD phase separation with non-cognate DNAs at these modest concentrations reinforces the idea that the failure of 6 µM DBD to support LLPS at 3 µM NANK despite robust LLPS at 1 or 2 µM NANK (Fig. 2i) results from the sequestering of DBD into canonical complexes that depopulate states in which ZnF1 is exposed. At 2:1 DBD:NANK, energetically favored canonical modes can be adopted without steric clashes, as in Fig. 4a, so few DBD will adopt either obligately or transiently

exposed binding modes. We reasoned that providing cognate sites in trans might therefore dissolve pre-formed condensate by sequestering DBD. We tested this by preparing 10 µM DBD with 3 µM NANK and allowing the mixture to undergo LLPS. Adding 0.5 equivalents of NANK causes rapid, total loss of the condensate (Fig. 5b, top). The initial and final states are consistent with the phase diagram (Fig. 2i), while the rapid dissolution shows that material readily exchanges between the aqueous phase and the enriched phase. This behavior and our stoichiometry-based explanation are similar to observations and the rationale for phase separation of tandem SH3 domains with a tandem substrate[47]. For a 3:1 DBD:FGF4 mixture, adding 0.5 equivalents of FGF4 DNA only modestly decreases the amount of condensate (Fig. 5b, bottom) because FGF4 has no high affinity KLF4 cognate sites.

**DNA sequence strongly influences LLPS threshold concentration**. To determine how the DNA sequence might influence the lowest concentration at which LLPS is observed (the threshold DNA concentration), we performed LLPS assays for the non-cognate 17-mer DNA SBE, for NANK, and for the CpG methylated substrate NANKm at a range of DBD concentrations (Fig. 5c). Even at 3 µM SBE, no LLPS is seen with 1.5 or 3.0 µM DBD; with 6.0 µM DBD, the threshold SBE concentration is 1.5 µM. For NANK, the thresholds for LLPS at 1.5, 3.0, and 6.0 µM DBD are 250, 125, and 125 nM (respectively); the 6.0 µM DBD threshold for NANK is more than 10-fold lower than that of SBE. For NANKm, the LLPS thresholds at 1.5, 3.0 and 6.0 µM DBD are 63, 31, and 16 nM (respectively); these thresholds are 4–8 fold lower than those for NANK, and the threshold for NANKm at 6.0 µM DBD is at least 90-fold lower than that of SBE. The degree of DNA interaction with DBD by EMSA correlates with DBD:DNA LLPS potential (Supplementary Fig. 6), and the very low threshold concentrations for NANKm indicate that condensation can be directed to high affinity KLF4 binding sites. We conclude that the DNA sequence can dramatically alter the propensity for DBD:DNA biomolecular condensation, and that CpG methylation of the *NANOG* promoter KLFB site (converting NANK to NANKm) strongly potentiates condensation.

**Zinc finger domain mutations attenuate DBD:DNA condensation**. If the lower threshold concentrations for NANKm compared to NANK are caused by tight DBD binding that accompanies CpG methylation (Fig. 3a), then residues that participate in KLF4 base-specific recognition (see Fig. 3c) should be important to LLPS. On the other hand, if the observed condensation depends on an unfolded DBD fraction, a different DBD surface, or a trace contaminant, then large-to-small mutations at the "specificity residues" of the C2H2 recognition code[39] should have no effect on LLPS. We, therefore, prepared DBD carrying a ZnF2 mutation (E476D, position 3) that weakens affinity for cognate KLF4 sites[48] and a ZnF3

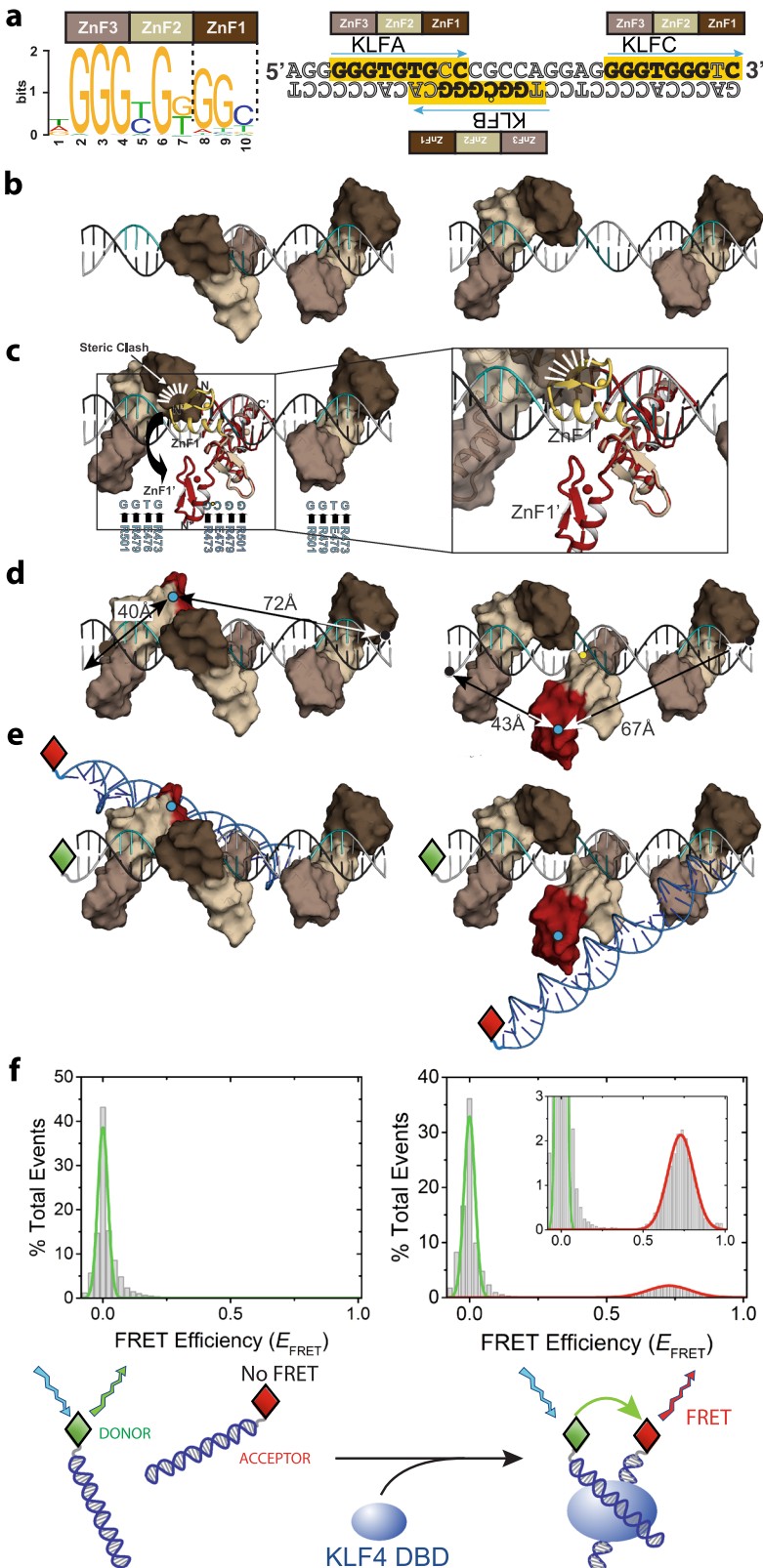

mutation (R501A, position −1) that weakens affinity by EMSA (Supplementary Fig. 6). The double mutant domain (DBD$^{E476D/R501A}$) shows decreased LLPS compared to wild type for all three of the tested DNAs (Fig. 5d). Even at 3.0 μM SBE, no LLPS is detected with 6.0 μM DBD$^{E476D/R501A}$. For NANK, no LLPS is detected for DBD$^{E476D/R501A}$ at 1.5 or 3.0 μM, though the threshold concentration at 6.0 μM is unaltered from wild type. For NANKm, no LLPS is

detected at 1.5 μM DBD$^{E476D/R501A}$, and the thresholds for LLPS at 3.0 and 6.0 μM are 8 fold higher for DBD$^{E476D/R501A}$ than for DBD. Two of the four non-cognate DNAs that condense robustly at 3 μM with 10 μM DBD (Fig. 5a) show no condensation at 3 μM with 10 μM DBD$^{E476D/R501A}$ (Supplementary Fig. 7). We conclude that the ZnF2 and ZnF3 surfaces that contact bases in cognate DNA are important for LLPS with both non-cognate and cognate DNAs.

**Fig. 4 Models and solution data for DBD interactions with NANK. a** Consensus KLF4 binding site from JASPAR[40] and schematic DBD contacts (left); mapping these sites and contacts onto the NANK sequence (right); in each 9 bp site (yellow), bases that match the consensus are in bold. **b** Posing DBD:NKA at cognate sites KLFB/KLFC (left) or KLFA/KLFC (right) on B-DNA NANK model generates no clashes. **c** B-DNA NANK model with canonical DBDs at KLFA and KLFC shown as surfaces and two poses for DBD at KLFB in cartoons. ZnF1 of DBD:NKA (yellow) clashes with DBD modeled at KLFA (top right) but an alternate ZnF1 pose (2wbs, red) does not. Positions of key residue:base contacts indicated in cyan. **d** Distances between modeled NANK 5′ ends and H446 Cα of ZnF1 sterically excluded from KLFA (left) or KLFB (right) favor 5′ labeling the coding strand. **e** Superimposing a second duplex on the excluded ZnFs suggests that 5′ labels could be used to detect DBD-mediated DNA bridging in solution by smFRET. **f** FRET efficiency ($E_{FRET}$) histograms of mixtures of 100 pM Alexa 488-labeled NANK and 500 pM Alexa 594-labeled NANK in the absence (left) or presence (right) of 1 μM KLF4 DBD were fit to Gaussian functions (donor emission, green; acceptor/FRET emission, red). Bottom schematic emphasizes that the atomic details of DNA:DBD:DNA bridging are not defined experimentally.

We then transfected cells with constructs carrying wild type, R501A mutant, or E476D/R501A double mutant KLF4-mTurq and assessed their distributions by microscopy. More puncta are seen for wild type than the mutants (Fig. 5e, top), but mutant expression levels are lower than wild type. Automating the identification of "punctate" cells (>5 puncta detected) and plotting cells by their average fluorescence values reveals that both mutant proteins can be expressed at higher levels than wild type without conferring a "punctate" phenotype (Fig. 5f). At levels between 5.0 and $7.5 \times 10^3$ arbitrary units, all cells expressing wild type fusions are classified as "punctate" but fewer than half of those expressing mutant proteins are so classified (Fig. 5f). Visual comparison of HEK 293T cells (Fig. 5e, bottom) or BJ fibroblasts (Supplementary Fig. 8) with equivalent average fluorescence confirms that the wild type construct supports a more punctate distribution than the point mutants. We conclude that at equivalent expression levels, both KLF4$^{R501A}$-mTurq and KLF4$^{E476D/R501A}$-mTurq undergo biomolecular condensation less readily than KLF4-mTurq. We infer that the DNA-contacting surfaces of ZnF2 and ZnF3 are therefore important to condensation mediated by full-length KLF4 in cells, and that the observed condensation is likely mediated by KLF4 molecules whose DNA binding domain is properly folded.

**KLF4 biomolecular condensates recruit SOX2 and OCT4**. We then tested whether SOX2 and OCT4, TFs that cooperate with KLF4 at promoters and enhancers[5,6,14,19], would co-localize to KLF4-mediated condensates by co-expressing KLF4-mTurq with OCT4-mCherry or SOX2-mCherry. OCT4-mCherry expressed alone shows a uniform nuclear distribution at low levels, with some tiny puncta at higher expression levels (Fig. 6a). SOX2-mCherry expressed alone shows distributions consistent with SOX2 acting as a bookmark for mitosis[49]: although usually uniform or showing tiny puncta, in some cells it highlights mitotic chromosomes (Fig. 6a). After co-transfection of vectors for TF-mCherry and KLF4-mTurq, only a fraction of cells express both tagged proteins. OCT4-mCherry co-localizes to KLF4-mTurq droplets and puncta (Fig. 6b, top) in all cells where both proteins are detected (n = 73, 2 biological replicates); OCT4-mCherry droplets are never seen in the absence of KLF4-mTurq. SOX2-mCherry co-localizes to KLF4 puncta and droplets in 74% of cells where both KLF4-mTurq and SOX2-mCherry fluorescence are detected (n = 39, 2 biological replicates); when SOX2-mCherry does not co-localize with KLF4-mTurq, its distribution resembles mitotic bookmarking (Fig. 6c, bottom). We conclude that the cellular KLF4 condensate can recruit OCT4 or SOX2.

To determine if the in vitro DBD:DNA condensed phase can recruit TFs, we labeled purified full-length OCT4 and SOX2 proteins with Alexa Fluor 647, mixed them with NANK (which lacks OCT4 or SOX2 cognate binding sites) with or without DBD, and monitored the mixtures by fluorescence microscopy. OCT4-AF647 or SOX2-AF647 mixed with NANK give homogeneous mixtures, but addition of DBD drives NANK into droplets that

co-localize with OCT4-AF647 or SOX2-AF647 (Fig. 6d). We conclude that the DBD-mediated biomolecular condensate can recruit TFs, perhaps through non-specific TF:NANK interactions.

We then assessed DBD behavior with a polynucleosome substrate consisting of a 5 kbp plasmid DNA with sites for 11 nucleosomes and at least 9 GGTG motifs (Active Motif, Inc.). DBD colocalizes to droplets with polynucleosomes (Fig. 6e), and droplets induced by mixing DBD with polynucleosomes recruit labeled OCT4 or SOX2 (Fig. 6f). DBD condenses with this substrate at low concentrations: 250 nM DBD induces droplets with 210 pM plasmid/nucleosome complex (0.4 ng DNA/μl, Fig. 6g, left). This might reflect DBD enhancing the intrinsic ability of polynucleosomes to phase separate[50]. The binding of KLF4 to DNA in nucleosomes[16] might mediate this enhancement or independently support condensation, but exposed plasmid DNA in this substrate might also drive condensation by recruiting many DBDs, giving it increased valency[51,52].

**KLF4 DBD condenses readily with long DNAs**. To see if longer DNAs containing NANK could condense readily without nucleosomes, we examined NP, a 404 bp *NANOG* promoter fragment (−379 to +25) that includes NANK and 6 additional GGTG sites. 250 nM DBD readily condenses with 2.5 nM NP, but not with NANK at the same DNA weight concentration (0.6 ng/μl, 32 nM) (Fig. 6g, center panels). The threshold concentration for NANK at 6 μM DBD is 125 nM (Fig. 5b), so NP condenses at 24-fold lower DBD levels and 4-fold lower DNA weight concentration (50-fold lower mole concentration) than NANK. 250 nM DBD condenses robustly with 0.6 ng/μl (130 pM) NPE, a 7.4 kbp linear DNA containing portions of the *NANOG* promoter and its −5 enhancer[53] and 93 GG(C/T)G sites (Fig. 6g, right). We conclude that long DNAs condense much more readily than short DNAs.

## Discussion

We propose that KLF4 organizes chromatin by forming condensates at genomic loci to which it is recruited in high numbers and then stabilizing the colocalization of such genomic sites when their KLF4:DNA condensates fuse during random diffusive collisions (Fig. 7). For the initial condensation, we expect that KLF4 would bind tightly to 6 bp on one DNA through ZnF2/ZnF3 but more weakly to another DNA through ZnF1, in a bridging mode, and that several KLF4 bound to one stretch of DNA would provide the valency needed to drive biomolecular condensation[51,52]. Cognate KLF4 sites[19], overlapped sites (Fig. 4a, c), and partial 6 bp sites[16] that might direct KLF4:DNA condensation at particular genomic loci will have their affinities modulated by CpG methylation and their accessibility influenced by nucleosomes and by other DNA binding proteins. KLF4:DNA condensation in vitro does not require IDR:IDR interactions, but in cells the KLF4 IDR may contribute to condensation (through homotypic interactions) or to recruitment of other factors

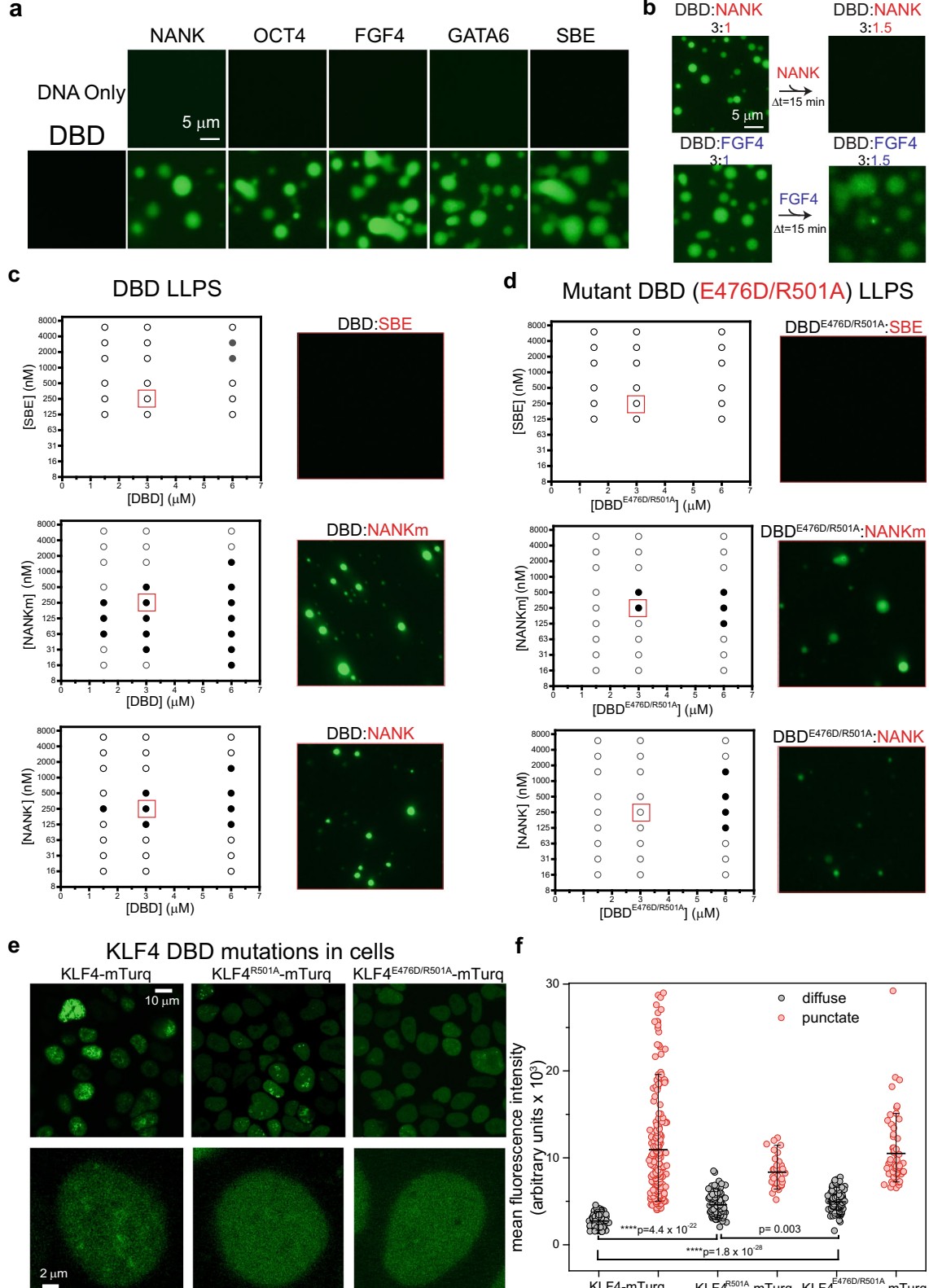

(through heterotypic interactions). Chromatin modifying machinery would be able to reinforce or reverse KLF4:DNA condensation by altering the accessibility or methylation states of KLF4 binding sites.

KLF4 is found at both repressive and activating loops in PSCs[15], indicating that contacts mediated by KLF4:DNA condensates are not sufficient to drive transcriptional activation.

Spatial colocalization of genomic elements by KLF4:DNA condensates combined with IDR-centric models for transcriptional activation[23–26,54,55] can explain many observed chromatin features. Promoters and enhancers associated with pluripotency are known to recruit KLF4 when they make long-range contacts[5,12–15]; we propose that KLF4 is condensed with DNA at these loci, helping to stabilize the observed long-range contacts

**Fig. 5 DNA affinity influences KLF4-mediated liquid condensation. a** Fluorescence microscopy images of 3 μM duplex DNA in TS buffer and 100 nM YOYO-1 with (bottom) or without (top) 10 μM DBD. Similar results were obtained for 2 replicates. **b** Fluorescence microscopy images of DBD:DNA mixtures at 3:1 and 3:1.5 ratios show that adding cognate DNA can reverse LLPS. Similar results were obtained for 2 replicates. **c** Mixing DBD and DNAs at different concentrations produces condensates (black circles) or homogeneous solutions (open circles) that define LLPS phase diagrams. Images were scored as condensate if the coefficient of variation was >0.2 and the mean fluorescent intensity was >0.4 arbitrary units. Values were determined using ImageJ from 2–3 independent replicates. Fluorescence images at right are taken at 3 μM DBD and 250 nM DNA, conditions boxed in red in the phase diagrams. **d** LLPS phase diagrams determined as in (**c**) but for double mutant DBD$^{E476D/R501A}$ with DNAs. **e** Fluorescence images of HEK 293T cells expressing KLF4-mTurq (left), KLF4$^{R501A}$-mTurq (middle) and KLF4$^{E476D/R501A}$-mTurq (right). Bottom row shows close-ups of single cells with the same mean fluorescence (~5000 AU). Similar results were obtained for 2 biological replicates. **f** HEK 293T cells expressing KLF4-mTurq variants classified as diffuse (gray, left) or punctate (red) are plotted by mean fluorescence intensity; the long horizontal bars mark the median, and the top and bottom bars on the vertical lines denote the 90-10 percentiles. Wildtype "diffuse" cells show lower fluorescence than those cells expressing KLF4$^{R501A}$-mTurq or KLF4$^{E476D/R501A}$-mTurq ($n = 112$ diffuse cells for each fusion across 2 biological replicates). Cells with at least 5 distinct puncta (0.5 μm spots with intensity center >1500 arbitrary units, determined using Imaris software) were classified as punctate. Statistical tests were performed using a two-sided Student's paired $t$-test. $Y$-axis limit was set to 30,000 for visualization purposes. Fields that yielded 112 diffuse cells also gave punctate cell counts of 209 (WT), 39 (KLF4$^{R501A}$-mTurq) and 64 (KLF4$^{E476D/R501A}$-mTurq) across 2 biological replicates.

(Fig. 7). Super-enhancers are larger than typical ESC enhancers, more enriched in KLF4, and able to recruit much higher levels of Mediator[14]. These properties can be explained by extensive KLF4:DNA condensates that bring together several enhancers, whose abilities to recruit transcription machinery through IDR:IDR interactions[24,26] would be increased by their mutual proximity and by recruitment of TFs to the KLF4:DNA condensate. The KLF4-mediated recruitment of histone demethylase JMJD3[56] or DNA demethylase TET2[57] may be influenced by KLF4:DNA condensation, and the KLF4-mediated recruitment of cohesin[13,56] may help to topologically link remote DNA segments held together by KLF4:DNA condensation.

KLF4 is functionally implicated at the *NANOG* promoter in somatic cell reprogramming[5–7,12–15,19]. We propose KLF4 binding and condensation as the first mechanistic steps in accessing the closed, highly methylated *NANOG* promoter during reprogramming. Silenced chromatin is compact, but KLF4 should diffuse into it readily because its folded ZnFs are small and its IDR is deformable. The human *NANOG* promoter has KLF4 cognate sites spaced by 15, 7, and 11 bp, so one of these sites must be partially exposed in nucleosomes, and KLF4 is known to bind to 6 bp partial sites in nucleosomal DNA[16]. KLF4 that binds to the CpG-methylated, nucleosome-wrapped *NANOG* promoter can recruit more KLF4 through condensation driven by its exposed ZnF1, and possibly through homotypic IDR:IDR interactions. When nucleosomal breathing motions expose DNA[58], locally tethered KLF4 ZnFs will occupy newly exposed major grooves and prevent rewrapping. The local KLF4:DNA condensate will recruit TFs OCT4 and SOX2, biasing their diffusive searches[59] to promoter sites within the condensate and further favoring nucleosome unwrapping; heterotypic IDR:IDR interactions between KLF4 and TFs could enhance recruitment.

Rising KLF4 levels early in reprogramming (Fig. 7a) will promote growth and fusion of KLF4:DNA condensates that help determine the long-range contacts made by the *NANOG* promoter (Fig. 7b–g). KLF4-enriched enhancers and promoters (Fig. 7c) that collide by random diffusion (Fig. 7d, e) will remain co-localized due to fusion of their KLF4:DNA condensates, within which KLF4 DNA bridging mediates a network of contacts among the key loci and nearby DNA (Fig. 7f, g). These steps driven by KLF4 expression could clear the way for recruitment of transcription machinery that initiates *NANOG* expression in mid-to-late stages of reprogramming[12]. A role for KLF4:DNA condensation in organizing chromatin can explain why an additional copy of KLF4 increases the efficiency of somatic cell reprogramming methods[11] and commercial kits[60] (CytoTune 2.0, Thermo Fisher), and why limiting KLF4 expression halts

reprogramming at distinct stages of epigenetic reset but increasing KLF4 levels drives partially reprogrammed cells to iPSCs[61].

DNA bridging by tandem $C_2H_2$ zinc fingers that we demonstrate here for KLF4 (Fig. 4f) could be widely implicated in chromatin structure and gene expression: the human genome contains more than 700 $C_2H_2$ ZnF proteins with four or more tandem ZnFs, having an average of 8.5 and as many as 30 ZnFs[62]. Many such proteins have ZnFs that are not needed to bind their DNA cognate sites and so might make bridging contacts; for instance, just three of the 11 tandem ZnFs in TZAP are sufficient to direct proteins to telomeres[63]. The TF GLIS1, which uses two of its five ZnFs to recognize targets[64], enhances reprogramming by OCT4/SOX2/KLF4[65]; if it were to make bridging contacts with its other three ZnFs, such contacts could be long-lived. ZnFs with unidentified functional roles are also common in proteins with repressive effects in chromatin: the repressor ZFP57 binds a methylated 6 bp motif in closed chromatin with two of its seven ZnFs[66], and the N-terminal ZnF of the mouse repressor protein ZFP568 does not contact target DNA[67]. The architectural protein CTCF, which interacts through its N-terminal domain with cohesin[68] and whose binding site polarity on DNA controls chromatin looping[69], makes sequence-specific contacts with different target DNAs but its terminal ZnFs (ZnF1, ZnF10, and ZnF11) do not contribute to binding target DNA[70]. Our demonstration that the KLF4 ZnF tandem array makes DNA-bridging contacts that mediate condensation suggests that other $C_2H_2$ tandem ZnF proteins may bridge DNA making transient or long-lived contacts that contribute to biological function.

## Methods

**Bacterial strains.** The *E. coli* strain DH5α (Thermo Fisher Scientific) was used for plasmid cloning and large-scale preparations of plasmid DNAs. The *E. coli* strain BL21 Star (DE3) (Thermo Fisher Scientific) was used for large-scale protein production.

**Mammalian cell lines.** The HEK 293T cell line (from ATCC, CRL-3216), Lenti-X 293T (from Takara Bio USA, TaKaRa Bio # 632180), and BJ fibroblasts (from ATCC, CRL-2522) were cultivated in Dulbecco's Modified Eagle Medium (DMEM, Corning) with 10% (v/v) fetal bovine serum (FBS, Corning) and 1X antibiotic-antimycotic solution (Corning). All cells used in this study tested negative for mycoplasm contamination.

**Construction of mammalian plasmids.** All generated constructs and mutations were confirmed by DNA sequencing (Eurofins Genomics). The pHRT-GFP-AH lentiviral transfer vector was generated from pHR-CMV-TetO2_3C-Avi-His6 (Addgene #113887) by replacing the DNA fragment corresponding to the 5′-Chicken RPTPs signal sequence-HRV 3C site-3′ with a DNA fragment corresponding to 5′-BamHI-KpnI-TEV cleavage site-eGFP-3′. The insert fragment was amplified from the plasmid encoding TEV-eGFP using the primers eGFP-F/eGFP-R; see Supplementary Table 1 for all primers. The vector fragment was amplified

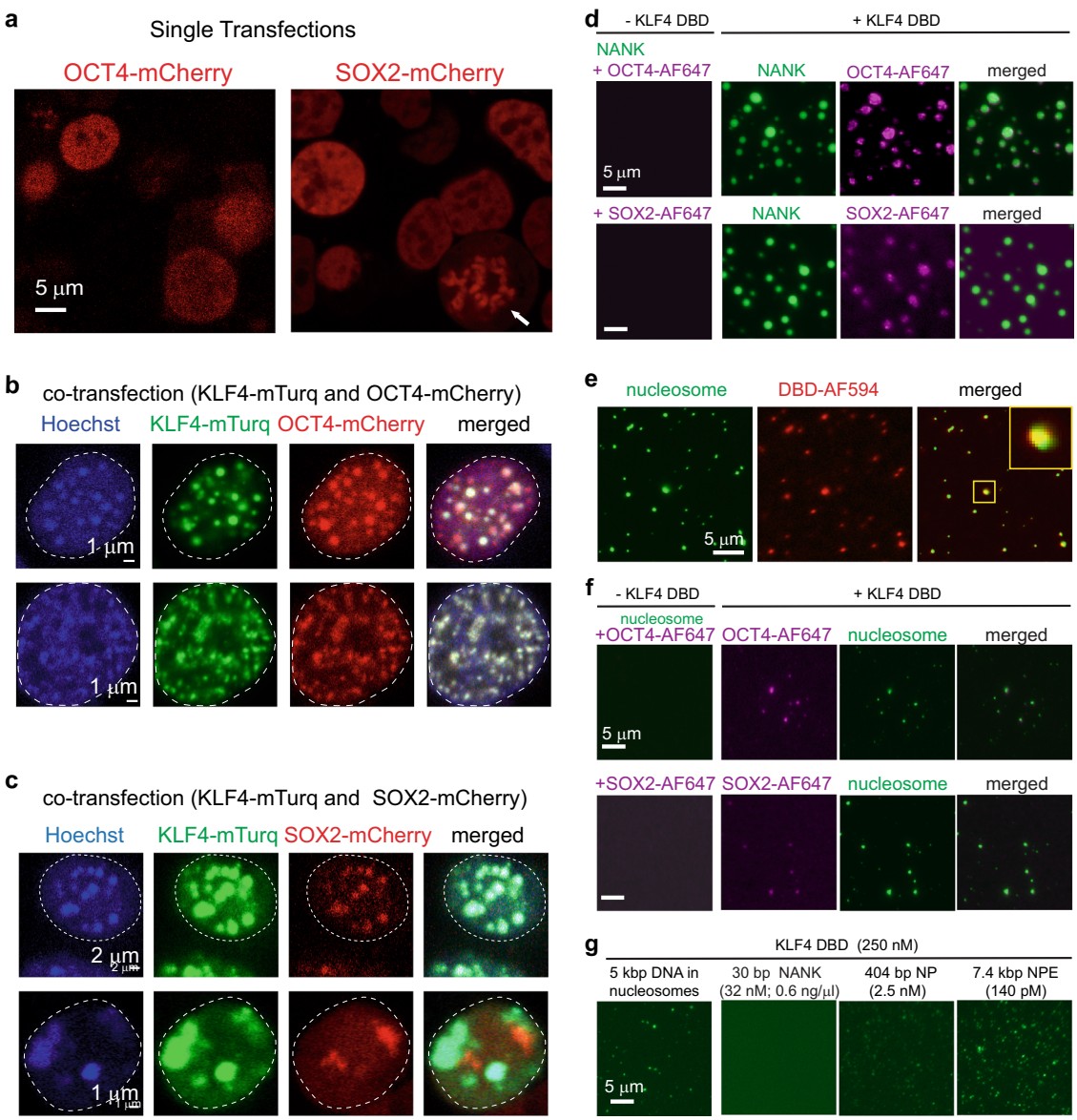

**Fig. 6 KLF4 condensates recruit TFs and form at low concentrations with long DNAs. a** Fluorescence microscopy images of HEK 293T cells expressing OCT4-mCherry (left) or SOX2-mCherry (right). Some SOX2-mCherry distributions suggest mitotic bookmarking[49] (white arrow). **b** Fluorescence microscopy images of HEK 293T cells. Screening ~1000 cells from 2 transfection replicates identified 73 cells that co-express OCT4-mCherry and KLF4-mTurq; all 73 show tags colocalized to droplets (examples top and bottom). **c** Fluorescence microscopy images of HEK 293T cells. Screening ~1000 cells from 2 transfection replicates identified 39 cells that co-express SOX2-mCherry and KLF4-mTurq; 29 cells show tags colocalized to droplets (example, top row) and 10 cells do not (example, bottom row). **d** Fluorescence microscopy images of 1.5 µM NANK DNA with 100 nM YOYO-1 and 50 nM OCT4-AF647 (top) or 70 nM SOX2-AF647 (bottom) without (left column) or with (right three columns) 9 µM DBD. Similar results were obtained for 2 replicates. **e** Fluorescence microscopy images of polynucleosomes (green; 20 ng DNA/µl, Active Motif, Inc., visualized with 100 nM YOYO-1) with 10 µM DBD (red; trace labeled 1:100 with DBD-AF594). Similar results were obtained for 2 replicates. **f** Fluorescence microscopy images of polynucleosomes (green; 11 ng DNA/µl, visualized with 100 nM YOYO-1) with OCT4-AF647 (50 nM; purple, top row) or SOX2-AF647 (70 nM; purple, bottom row) alone (left) or with 1 µM DBD (three right panels). Similar results were obtained for 2 replicates. **g** Longer DNAs undergo LLPS with DBD at low concentrations. Fluorescence images of DBD with 30 bp NANK, 404 bp NP (*NANOG* promoter), 7.4 kbp NPE (*NANOG* promoter enhancer) and 5 kbp plasmid DNA in nucleosomes (left to right panels, respectively). Conditions: DBD (250 nM) mixed with different DNA (mass equivalent of 0.6 ng/µl; 32 nM NANK or 2.5 nM NP or 140 pM NPE or 210 pM plasmid DNA concentration in nucleosomes) in TS buffer with 100 nM YOYO-1. Similar results were obtained for 2 replicates.

from pHR-CMV-TetO2_3C-Avi-His6 using the primers pHRT-1F/pHRT-1R. The insert and vector fragments were ligated together using Gibson Assembly Master Mix (NEB) according to the manufacturer's protocol.

Lentiviral vectors pHRT-mTu-AH and pHRT-mCh-AH were generated by replacing the eGFP gene in pHRT-GFP-AH with mTurquoise2 (mTu) and mCherry (mCh), respectively. The mTu insert was amplified from pmTurquoise2-Tubulin (Addgene #36202) using the primers mTu-1F/mTu-1R. The mTu gene has the A206K mutation to ensure obligate-monomer state[71]. The mCh insert was amplified from pBRY-nuclear mCherry-IRES-PURO (Addgene #52409) using the primers mTu-1F/mTu-1R. The vector fragment was amplified from pHRT-GFP-

AH using the primers pHRT-1F/pHRT-2R. The insert and vector fragments were ligated together using Gibson Assembly Master Mix.

Vector pHRT-KLF4-mTu-AH was constructed to express KLF4 with a C-terminal TEV cleavage site (mTurquoise2-Avi-His6; mTu-AH) in a lentiviral expression system. The KLF4 insert was amplified from the plasmid encoding KLF4 (GeneArt) using primers KLF4-1F/KLF4-1. The insert fragment was digested with *Bam*HI/*Kpn*I and ligated into *Bam*HI/*Kpn*I-digested pHRT-mTu-AH lentiviral transfer vector using T4 DNA ligase (Promega).

pHRT-KLF4(2-417)-mTu-AH and pHRT-KLF4(418-513)-mTu-AH were constructed to express labeled KLF4 deletion constructs. The KLF4 coding regions

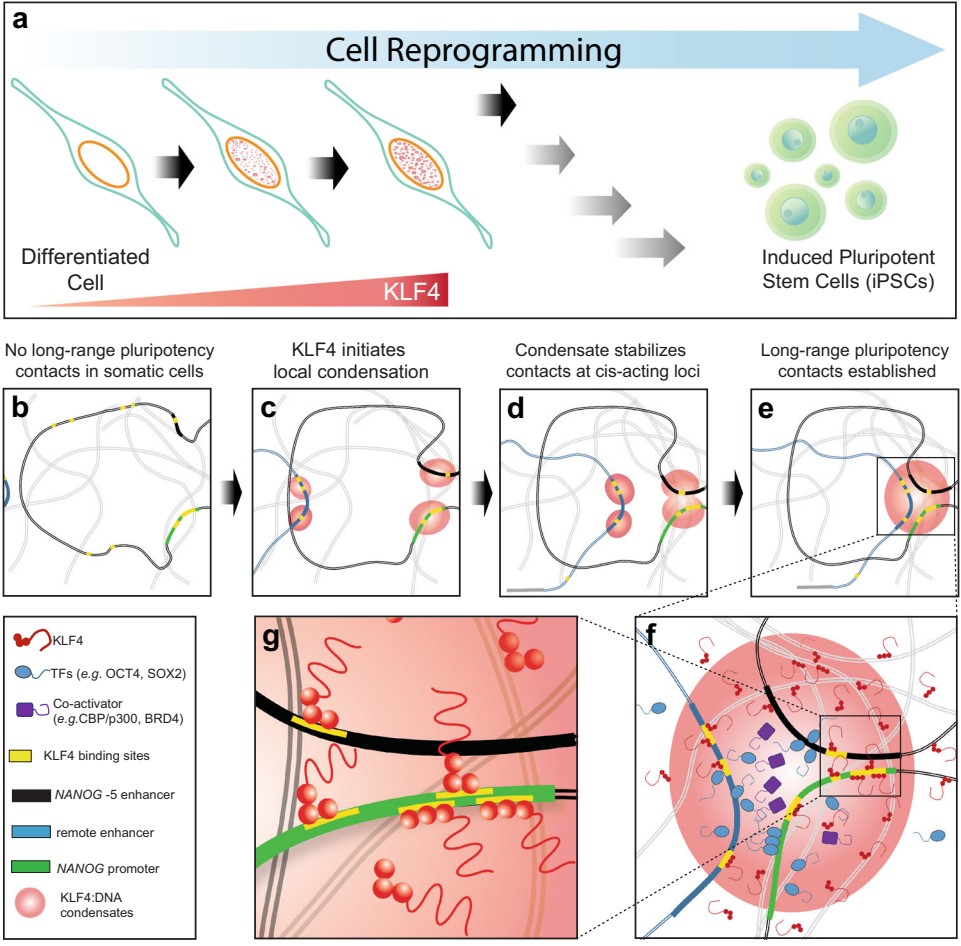

**Fig. 7 KLF4:DNA condensation as an organizer of chromatin. a** KLF4 levels rise early in reprogramming, leading to KLF4 biomolecular condensation. **b–g** Proposed roles for KLF4:DNA condensates in chromatin reorganization during reprogramming. **b** Pluripotency-related sites in closed chromatin of somatic cells lack KLF4. **c** Sites will recruit KLF4 and form local KLF4:DNA condensates as expression levels rise. **d** Random diffusive encounters of cis-acting elements leads to fusion of their condensates and persistent colocalization. **e** Loci on different chromosomes can also be co-localized in KLF4:DNA condensates. **f, g** Within the condensate, KLF4 makes DNA bridging contacts at cognate 9 bp KLF4 sites (yellow), especially where such sites overlap, but perhaps also at 6 bp partial sites or non-cognate sites. The KLF4:DNA condensate enriches TFs relative to solution, helping to saturate binding sites for TFs OCT4 and SOX2 and to recruit co-activators such as CBP/p300 to activate NANOG expression and gain access to pluripotency.

corresponding to the intrinsically disordered region (IDR, residues 2-417) or the DNA binding domain (DBD, residues 418–513) were amplified from the plasmid encoding human KLF4 gene (GeneArt) using primer sets KLF4-2F/KLF4-2R or KLF4-3F/KLF4-3R, respectively, and ligated into BamHI/KpnI-digested pHRT-mTu-AH lentiviral transfer vector using Gibson Assembly Master Mix.

pHRT-KLF4_R501A-mTu-AH. To construct the KLF4 single mutant (R501A), two DNA fragments were amplified from pHRT-KLF4-mTu-AH using the primer sets (KLF4-4F/KLF4-4R and KLF4-5F/KLF4-5R, respectively) and ligated together using Gibson Assembly Master Mix.

To construct the KLF4 double mutant (E476D/R501A) expression vector pHRT-KLF4_E476D_R501A-mTu-AH, two DNA fragments were amplified from pHRT-KLF4_R501A-mTu-AH using the primer sets KLF4-4F/KLF4-6R and KLF4-7F/KLF5R, respectively, and ligated together using Gibson Assembly Master Mix.

To express fluorescently labeled OCT4 in the lentiviral expression system, the pHRT-OCT4-GFP-AH plasmid was built by amplifying the OCT4 gene from pGEX4T-1_WT_OCT4 (Addgene #40633) using the primers OCT4-1F/OCT4-1R and ligating into BamHI/KpnI-digested pHRT-GFP-AH using Gibson Assembly Master Mix. The pHRT-OCT4-mCh-AH construct for expression of the C-terminal TEV cleavage site-mCherry-Avi-His6 (mCh-AH) fused OCT4 was built by amplifying the OCT4 gene in pHRT-OCT4-GFP-AH using the primers OCT4-2F/OCT4-2R and ligating into EcoRI/KpnI-digested pHRT-mCh-AH using Gibson Assembly Master Mix.

To construct pHR-SOX2-mCh-Cry2olig, the SOX2 gene (insert) was amplified from the SOX2 gene in pEP4 E02S EN2L (gift from James Thomson, Addgene #20922) using the primers SOX2-1F/SOX2-1R. The vector fragment was amplified from pHR-mCh-Cry2olig (gift from Clifford Brangwynne, Addgene #101222) using the primers pHRT-3F/pHRT-3R. The insert and vector fragments were ligated together using Gibson Assembly Master Mix.

**Construction of bacterial plasmids.** To express the N-terminal streptavidin-binding Nano tag-His6-TEV cleavage site (NH6t) fused KLF4 DBD in the bacterial expression system, the KLF4 DBD gene (insert) was amplified with the plasmid encoding KLF4 (GeneArt) using primer sets (KLF4-8F/KLF4-8R). The vector fragment was amplified from pET15Nano6HT-SMAD1 (DNASU) using primer sets (pET15-1F/pET15-1R). The insert and vector fragments were ligated together using Gibson Assembly Master Mix to give pET15-NH6t-KLF4(418–513). KLF4 DBD mutations E476D and R501A were introduced into plasmid pET15-NH6t-KLF4(418–513) using the QuikChange Multi Site-Directed Mutagenesis Kit (Agilent) and the primers KLF4-9F/KLF4-10F according to the manufacturer's protocol. Product pET15-NH6t-KLF4(418–513)_E476D_R501A was verified by DNA sequencing; residue numbering follows the human gene product.

**Lentiviral transfection and transduction.** Recombinant lentiviruses were produced by co-transfection of Lenti-X 293T cells ($1 \times 10^6$ cells in gelatin-coated 10 cm cell culture dish) with lentiviral transfer construct (1.2 pmol), psPAX2 packaging plasmid (1.2 pmol; Addgene # 12260), and pMD.2G envelope plasmid (0.7 pmol; Addgene #12259) using the Lipofectamine 3000 Transfection Reagent (Thermo Fisher Scientific). Transfection was performed according to the manufacturer's optimized protocols. Lentiviruses were harvested 3 days post-transfection, filtered through 0.45 μm-pore size PES filters, and concentrated 100 times using Lenti Concentrator (OriGene). The lentiviral titer concentration, determined by Lentivirus Titer Kit HIV-1 p23 Elisa Assay (OriGene), was ~$2 \times 10^8$ TU/mL.

Lentiviral transduction was carried out with $1 \times 10^5$ host cells in 24-well cell culture plate and 10 multiplicities of infection (MOIs) of lentivirus in the presence of 8 μg/mL of polybrene. Lentivirus was removed after overnight incubation, and fresh cell culture media was added. Two days post-transduction, expression of the

fluorescent proteins (eGFP, mTurquoise2, and mCherry) were verified using an EVOS fluorescence microscope (Thermo Fisher Scientific). Transduced cells were passaged in culture as bulk preparation for functional assays.

**Purification of KLF4 DBD (418–513; DBD) WT and double mutant (E476D/ R501A).** Either pET15-NH6t-KLF4(418–513) or pET15-NH6t-KLF4(418–513) _E476D_R501A were transformed into *E. coli* BL21 Star competent cells (Novagen, Merck KGaA, Darmstadt, Germany). Transformed cells were grown at 37 °C in Terrific Broth media containing 100 μg/mL carbenicillin antibiotic until optical density at 600 nm (OD$_{600}$) reached 0.6. The culture was then transferred to an 18 °C incubator shaking at 250 rpm until OD$_{600}$ reached 0.8–1.0. Protein expression was induced with 1 mM IPTG, followed by overnight growth at 18 °C with shaking at 250 rpm. Cells from the overnight culture were harvested by centrifugation at 7900 × g. Both WT and double mutant (E476D/R501A) KLF4 DBD were purified with the same procedure. The cell culture pellets were resuspended in denaturing lysis buffer (6 M urea, 1 mM 2-mercaptoethanol, 0.5 M NaCl, 109 mM sodium phosphate, pH 8). The resuspended pellets were lysed using a cell homogenizer (Avestin, Ottawa, Canada), with the soluble fraction separated from the cell debris by centrifugation at 38,700 × g. Lysate containing the soluble fraction was filtered using a 0.25 μm filter (Corning). The His-tagged fusion protein was purified from the crude protein mixture by immobilized metal-affinity chromatography (IMAC) using batch/gravity method. The lysate was applied to a pre-equilibrated 5 mL HisPur cobalt resin (Thermo Fisher Scientific) followed by extensive washing (20–50 column volumes). The protein was eluted using elution buffer (denaturing lysis buffer + 200 mM imidazole). The eluted protein was combined with 3 volumes of 0.1% (v/v) TFA, and acidified to ~pH 3. The soluble fraction was further purified by reverse-phase HPLC using Zorbax 300SB C3 column (Agilent Technologies) and a 20–60% ACN gradient (Buffer A: dH$_2$O with 0.1% (v/v) TFA; Buffer B: ACN with 0.1% (v/v) TFA). Pure fractions (by SDS-PAGE analysis) were then combined and dialyzed against deionized distilled water (dH$_2$O). Afterward, the protein solution adjusted to ~ pH 6-7 with 1 M Tris, pH 8 (final concentration ~10-20 mM). TEV protease was then added (1 mg TEV: 25 mg protein), and the sample incubated overnight at 4 °C with rotation. The cleaved, untagged proteins were subsequently re-purified by HPLC (as described above), followed by dialysis with dH$_2$O and flash freezing using liquid N$_2$ prior to storage at −80 °C. CD spectroscopy (Aviv, Lakewood, NJ) was utilized to verify proper protein refolding, monitoring the transition from unfolded to folded state induced by changes in pH and the incremental addition of ZnSO$_4$. For crystallization and subsequent experiments, refolding was performed by addition of 3.3 molar equivalents of ZnSO$_4$ followed by buffer dilution using either 10 mM Tris, pH 8, or 1× TBS (140 mM NaCl, 25 mM Tris, pH 7.4) buffers.

**Purification of mTurquoise2.** For expression and purification of mTurquoise2 (mTurq) from bacterial cells, the pET15-mTu-AH plasmid was transformed into *E. coli* BL21 Star (DE3) chemically competent cells, which were then grown at 37 °C in Terrific Broth media with 100 μg/mL carbenicillin. When the OD$_{600}$ reached 1.0 to 1.5, protein expression was induced with 1 mM IPTG. The cells were incubated overnight with shaking at 18 °C and harvested by centrifugation. Pellets were resuspended and lysed in 1–2 mL RIPA2 lysis buffer solutions using a handheld sonicator operating at 30% power for three cycles of 60 s on, 60 s off. RIPA2 lysis buffer consists of 1× PBS (1.8 mM KH$_2$PO$_4$, 10 mM Na$_2$HPO$_4$, 2.7 mM KCl, 137 mM NaCl), 0.5% Triton X-100, and 0.1% sodium deoxycholate. The fluorescent protein was purified using batch/gravity immobilized metal-affinity chromatography (IMAC). The beads were extensively washed with 50 column volumes of RIPA2 buffer plus 500 mM NaCl, and the protein was eluted with 200 mM imidazole. The eluate was diluted and passed through Q Sepharose beads. The protein was eluted with 500 mM NaCl, concentrated, and exchanged into a new buffer with 2 mM TCEP, 10% glycerol, 500 mM NaCl, 25 mM Tris, pH 7.5.

**Purification of full-length (FL) OCT4 and SOX2.** *E. coli* BL21 Star competent cells (Novagen) were transformed with pGEX4T-1_WT_OCT4 (GST-OCT4 fusion) or pET302-GB1-SOX2 (His-tag protein G1 (h6GB1)-SOX2 fusion) expression plasmids. Protein expression was conducted as described above for KLF4 DBD. For h6GB1-SOX2, the final harvested cell culture pellet was resuspended in denaturing lysis buffer (8 M urea, 850 mM NaCl, 50 mM Tris, pH 8), lysed, and centrifuged. The supernatant was passed through an IMAC column with Co$^{2+}$ resin. After 20 column volumes of washing with the lysis buffer, the protein was eluted using the same buffer plus 200 mM imidazole. The eluted protein was concentrated, diluted six-fold with refolding buffer (1× PBS plus 500 mM NaCl, 5% (v/v) glycerol, and 0.1% (v/v) Tween-20). The h6GB1 fusion tags from h6GB1-SOX2 proteins were cleaved (1:20 TEV:protein w/w ratio) overnight at 4 °C. Co$^{2+}$ resin was used to remove h6GB1 and uncleaved proteins; Q Sepharose beads (GE Healthcare) were subsequently used to remove excess DNA. The flow through was mixed with TFA to a final concentration of 0.2% TFA and purified by C3 reverse phase HPLC using the procedure and gradient described above for the KLF4 DBD constructs. Purified fractions were lyophilized using Virtis BenchTop Pro (SP Scientific) and stored at −80 °C. Full length GST-OCT4 fusion proteins were first purified using standard non-denaturing GST purification methods. Briefly, cells were lysed in 1X PBS with 0.1% Triton X-100 and 5 mM DTT. The supernatant

was bound to GST Sepharose beads (GE Healthcare), the beads were washed extensively, and the protein was eluted with 50 mM Tris, 10 mM GSH, pH 8. The eluate was dialyzed and cleaved overnight in 1× PBS and TEV protease (1:20 TEV:protein w/w ratio). The solution was passed through GST Sepharose beads to remove GST tag and any uncleaved fusion proteins. The flow through and precipitates from dialysis, which contained cleaved OCT4, were dissolved in 6 M guanidine hydrochloride (GdnHCl) and purified by C3 reverse phase HPLC (as described above). Purified OCT4 fractions were lyophilized and stored at −80 °C. Refolded OCT4 and SOX2 are functionally active (assayed by EMSA).

**Fluorescent labeling of OCT4 and SOX2.** OCT4 and SOX2 were labeled with Alexa Fluor 647 (AF647) maleimide (Thermo Fisher Scientific) using standard methods described previously[45]. Briefly, the proteins were dissolved in 6 M GdnHCl, 20 mM Tris pH 8, mixed with 3–4 molar excess of Alexa Fluor 647 (AF647) maleimide dyes, and incubated for 1 h at RT. Samples were then mixed with 3-fold excess of 0.1% TFA/dH$_2$O and purified by reverse phase HPLC. Purified fluorescent labeled samples were lyophilized and stored at −80 °C. SOX2-AF647 and OCT4-AF647 protein samples had 45% and 135% fluorescent labeling efficiency, respectively. For colocalization experiments (Fig. 6d, f), SOX2-AF647 was dissolved in 6 M GdnHCl and diluted ~200× in 10 mM sodium phosphate buffer, pH 8. OCT4-AF647 protein was dissolved in 6 M GdnHCl and then buffer exchanged with NAP-5 columns (GE Healthcare) to a final concentration of ~500 nM in 10 mM sodium phosphate buffer, pH 8. Samples were snap frozen for storage at −80 °C before use.

**Fluorescent labeling of NANK.** NANK DNA oligos with 5′ amino modified C6 (IDT) were purified by ethanol precipitation and labeled with a 10-fold molar excess of dye (Alexa Fluor 488 or 594 NHS ester; Invitrogen). NANK has 30% (Alexa Fluor 488) and 50% (Alexa Fluor 594) fluorescent labeling efficiency. The labeling reactions were performed at 30 °C with 30–60 min incubation. The Alexa Fluor 488- or 594-labeled NANK were then ethanol precipitated; the collected DNA pellets were dissolved in 0.1 M triethylammonium acetate at pH 7. Excess unconjugated dyes were removed by passing two times over NAP-5 columns (GE Healthcare).

**Protein and DNA concentration determination.** DBD protein concentration was calculated based on the UV absorbance extinction coefficient at 280 nm of 22,190 M$^{-1}$ cm$^{-1}$ (based on Tyr and Trp absorbance[72]). All DNA oligonucleotides (Supplementary Table 2) for crystallization, LLPS and EMSA experiments were obtained from (Integrated DNA Technologies, Inc., Coralville, IA). Unlabeled duplex DNA for unmethylated and methylated DNA were calculated using the extinction coefficient of the single-strand DNA (IDT) and the formula[73] that accounts for the hypochromicity (h): $\{\varepsilon_{ds,260nm} = (\varepsilon_{ss,260nm} + \varepsilon_{reverse\ complement,260nm}) \times (1 - h)\}$ and $\{h = (0.059 \times f_{GC}) + (0.287 \times f_{AT})\}$, where $f_{GC}$ and $f_{AT}$ are fractions of GC and AT, respectively. Fluorescent DNA concentration was measured using the extinction coefficient of the Alexa Fluor 647 dye. Fluorescent labeling efficiencies were calculated using the corrected extinction coefficients based on the manufacturer's protocol (Invitrogen).

**Preparation of 404 bp NP (*NANOG* promoter) and 7.4 kbp NPE (*NANOG* promoter enhancer).** The human *NANOG* promoter was amplified from pNanog-Luc (Addgene #25900) using the forward primer hNan-F2 (or -F1) and the reverse primer hNan-R1 (or -R2); the primers in parentheses are fluorescently labeled versions of the listed primers; see Supplementary Table 2. The 404 bp PCR fragments were purified using QIAquik Gel Extraction Kit (Qiagen). The pGL-NanogP-5E minus plasmid[53] containing the mouse 1535 bp *NANOG* promoter and 1337 bp enhancer (−5 kbp from *NANOG* promoter) was digested with *Pvu*I (NEB) to linearize the plasmid. The DNA fragment containing *NANOG* promoter and enhancer was purified using QIAquik Gel Extraction Kit (Qiagen).

**Electrophoretic mobility shift assay (EMSA).** The binding reactions for the EMSA consisted of 1× EMSA buffer (0.01 mg/ml BSA, 0.1 mM DTT and 0.05 mM TCEP, 5% glycerol, 50 mM NaCl, 20 mM Tris pH 8) and unlabeled (50 nM–15 μM) protein (see figure legends for the exact protein and DNA concentration, and buffer conditions). Protein concentrations were prepared by 2-fold serial dilution. Samples were loaded onto either 10%, 12% or 4–15% pre-cast Mini-PROTEAN Tris-Glycine gel (TG; Bio-Rad) and electrophoresed for 25–45 min at 120 mV 4 °C in 1x TG buffer (Bio-Rad). EMSA experiments using unlabeled DNAs were stained with EtBr or Sybr™ Green for 20 min prior to imaging. The gels were then imaged using ChemiDoc with the appropriate filters and analyzed through the Image Lab software (Bio-Rad). EMSAs were performed with 2–3 independent replicates.

**Crystallization and X-ray data collection.** The KLF4 DBD:NKA complex was crystallized by hanging drop vapor diffusion method at 20 °C. Purified and refolded human KLF4 (418–513) in 0.5 mM DTT, 20 mM Tris-HCl, pH 8.0, and 3.3 molar equivalents of ZnSO$_4$ was mixed with 1.2 molar excess of dodecameric DNA (12-mer: 5′-AGG GGG TGT GCC-3′). Crystals of KLF4 DBD:NKA were obtained by mixing equal volumes of KLF4 DBD:NKA complex (40 mg/mL total

macromolecule) with 0.2 M sodium iodide (pH 7.0) and 20% w/v polyethylene glycol 3,350 reservoir solution. Single crystal X-ray diffraction data were collected at 100 K on the Beam Line 5.0.2 Advanced Light Source (UC Berkeley, USA) at wavelength (λ) = 1.00 Å, using an ADSC Q210 CCD detector. The collected data were integrated and scaled using iMosflm and SCALA, respectively[74,75].

**Structure solution and refinement**. The crystal structure of KLF4 DBD:NKA was determined by molecular replacement method using Phaser[76]. A prior crystal structure of KLF4 DNA binding domain (PDB ID: 2WBS) was used as search model. A unique solution was obtained for one molecule in the asymmetric unit. The dodecameric DNA was traced and fitted manually into electron density. The final model was obtained by iterative cycles of manual rebuilding using Coot[77] and refinement using phenix.refine[78]. PyMOL visualization program (https://pymol.org) was used for all the structural analyses and preparation of figures. The statistics for data collection and refinement are summarized in Supplementary Table 1. Residue numbering follows the human gene product; previous structures with identical ZnF sequences have been numbered according to the mouse gene product.

**In vitro LLPS microscopy imaging**. Monitoring for the presence/absence of LLPS droplets was performed at room temperature using EVOS fluorescence imaging system (Thermo Fisher Scientific) with bright field and/or necessary filters (CFP (mTurquoise2), GFP (YOYO-1, AF488), Texas Red (AF594), Cy5 (AF647)). For a set of experiments, the same light power and exposure time was used. Conditions for each set of experiments are detailed in the figure legends. To construct LLPS diagrams, various concentrations of KLF4 DBD and DNAs (cognate and non-cognate DNAs, see Fig. 2 for sequences) were prepared with either of the following buffers: TS buffer (70 mM NaCl, 12.5 mM Tris, pH 7.4; Figs. 2i, 5) or TS2 Buffer (140 mM NaCl, 25 mM Tris, pH 7.4; Supplementary Fig. 3). 100 nM YOYO-1 was added to samples in which the DNA was to be imaged by fluorescence microscopy. Specific conditions for the experiments are in the figure legends. LLPS diagrams were based on images obtained after 30 min of incubation. To assess colocalization of KLF4 DBD:NANK droplets and full length OCT4 or SOX2, KLF4 DBD (9 μM) was mixed with NANK DNA (1.5 μM) and either OCT4-AF647 (95 nM) or SOX2-AF647 (140 nM) in TS buffer. Samples were incubated for 30 min to 1 h prior to imaging. Experiments were performed in 2–3 independent replicates. To assess colocalization of KLF4 DBD with recombinant polynucleosomes purchased from Active Motif, as in Fig. 6e, the commercial polynucleosomes (H3.1; 20 μg protein + 24 μg 5 kbp plasmid DNA; 0.55 μg/μl) in 10 mM Tris-HCl, pH 8.0, 1 mM EDTA, 2 mM DTT, 20% glycerol were diluted (final concentration is 20 ng/μl) in TS buffer and mixed with KLF4 DBD (10 μM). To assess colocalization of KLF4 DBD, polynucleosomes, and OCT4 or SOX2, KLF4 DBD (1 μM) was mixed with commercial polynucleosomes (11 ng/μl) and OCT4-AF647 (50 nM) or SOX2-AF647 (70 nM) in TS buffer. Samples were incubated for 30 min to 1 hr prior to imaging. Experiments were performed in 2–3 independent replicates.

**Fluorescence live cell confocal imaging**. Fluorescence imaging of live cells (HEK 293 T cells and BJ fibroblasts plated on polyD-lysine coated 35 mm Ibidi μ-dish transduced with different plasmid constructs) was performed 2–3 days after lentiviral transduction using EVOS fluorescence imaging system (Thermo Fisher Scientific) or LSM780 and LSM880 laser-scanning confocal microscope system (Zeiss, Oberkochen, Germany) at 37 °C and 5% CO₂ with a ×60 oil objective. Images were analyzed using Fiji (ImageJ 1.52c), Zen 2.3 (Zeiss, Oberkochen, Germany) and Imaris v9.2 (Zurich, Switzerland) microscopy image analysis software. Images were taken at 3–5 different field locations for each biological replicate.

**Fluorescence recovery after photo-bleaching (FRAP) imaging in cells and in vitro**. FRAP imaging of KLF4-mTurq droplets and puncta/clusters in HEK 293T or BJ fibroblast cells (2–3 days after lentiviral transduction) were performed using a Zeiss LSM780 and LSM880 laser-scanning confocal microscope system at 37 °C and 5% CO₂. Different nuclear region of interest (ROI) spots (~0.5–2 μm diameter) were selected, and reference ROIs were drawn in adjacent regions (within the cell). Following 2–3 baseline images, ROIs were bleached for 50-200 iterations at 100% laser power (458 nm and 488 nm), and were imaged for up to 2–4 min post-bleaching for fluorescence recovery. FRAP recovery curves were corrected for background photobleaching (reference ROI in a separate droplet) and normalized against pre-bleach intensity values. FRAP data are fitted with an exponential function in the software Origin (Fig. 1c). FRAP imaging of LLPS droplets in vitro was achieved for LLPS droplets prepared by mixing trace-labeled KLF4 DBD (9 μM unlabeled DBD, 50 nM DBD-AF594) with trace-labeled NANK DNA (1.5 μM unlabeled NANK, 180 nM NANK-AF488). After ~1 h sample incubation, FRAP imaging was performed on droplets that had fused and settled close to the imaging surface. Using Zeiss LSM780 (with ×60 objective), different regions (~1 μm diameter ROI) were bleached with 100% power (488 nm) and 90% power (594 nm) for 100 iterations. Pre- and post-bleaching images (simultaneous 488 and 594 nm channels) were collected for ~15 min with 5 s intervals. After background subtraction (reference ROI in separate droplet) and normalization, the FRAP recovery curves (means and standard deviations) were plotted in the software Origin (Fig. 2g).

**Single-molecule Förster resonance energy transfer (smFRET)**. KLF4 DBD and NANK DNA binding interactions were monitored by single-molecule spectroscopy using a custom-built Alba confocal laser microscopy system (ISS, Champaign, Illinois). smFRET measurements were conducted in TS buffer (70 mM NaCl, 12.5 mM Tris, pH 7.4) at room temperature (21.5 ± 1 °C) by mixing 100 pM NANK 5′-labeled with Alexa Fluor 488 (FRET donor; Thermo Fisher Scientific) and 500 pM NANK 5′-labeled with Alexa Fluor 594 (FRET acceptor; Thermo Fisher Scientific), with or without 1 μM KLF4 DBD. Measurements were performed with 2 independent replicates. Freely diffusing FRET samples were excited with a 488-nm laser (ISS; ~115 μW). Fluorescence emission was split into donor-acceptor fluorescence by a 605-nm long pass beam splitter dichroic, and donor and acceptor signals were further filtered using 535/50-nm and 641/75-nm bandpass emission filters, respectively. Emission was detected using SPCM-ARQH-16 Avalanche photodiode detectors (Excelitas Technologies Corp., Waltham, MA). Data acquisition and FRET efficiency analysis were performed using VistaVision (64) 4.2.220.0 (ISS), correcting for acceptor emission due to direct excitation (1%) and fluorescence bleed-through of donor emission into the acceptor channel (5%), applying a binning time of 500 μs. There were 40,335 and 33,160 events collected for DNA samples without DBD and with DBD, respectively (Fig. 4). smFRET histograms were fitted to Gaussian functions using OriginPro 2020 (OriginLab, Northampton, MA, USA). FRET efficiencies ($E_{FRET}$) were calculated (using a value of unity for γ) from the corrected donor ($I_D$) and acceptor ($I_A$) fluorescence intensities as given by:

$$E_{FRET} = \frac{I_A}{I_A + \gamma I_D}$$

**LLPS quantification and statistical analysis**. To construct phase diagrams, a matrix of different nucleic acid and protein concentrations were mixed and incubated for 30 min. Images (fixed size of 153 × 114.7 μm) were collected at the same focal plane using EVOS microscopy system. The mean fluorescent intensities and standard deviation of the *.tif images were determined by the ImageJ software. Data from 2–3 independent replicates were averaged; the coefficient of variation (CV) is determined by the standard deviation divided by the mean. Positive phase separation for a particular condition is determined by CV > 0.2 and mean fluorescent intensity >4 arb. units (Figs. 2 and 5).

**Quantification of fluorescence intensities and puncta in cells**. Statistical tests (student's paired t-test) performed on experimental data and their representations are performed using Origin and noted in the figure legends. Puncta/droplet identification was determined through the Spots Algorithm in Imaris software v.9.2. Only spots that are localized in the nucleus, >500 nm in diameter and >1500 arb. units center intensity were chosen. The mean fluorescent intensities were determined by the Imaris software for the HEK 293T cells (Figs. 2 and 5) and ImageJ software for BJ fibroblasts (Supplementary Fig. 8).

**Nuclear concentration determination of KLF4-mTurq**. Nuclear concentrations of transiently expressed KLF4-mTurq were determined using a calibration plot of the fluorescence intensity/exposure time versus concentration of purified mTurquoise2 protein (Supplementary Fig. 2b). Using an EVOS fluorescence microscope, ×60 objective and CFP filter (Thermo Fisher Scientific), HEK 293T cells expressing KLF4-mTurq plated on 35 mm Ibidi μ-dish were imaged using 30% power, 15 ms exposure time. The nuclei boundaries were manually drawn in ImageJ and the mean fluorescence intensities quantified. The calibration plot was linearly fitted using Origin.

**Reporting summary**. Further information on research design is available in the Nature Research Reporting Summary linked to this article.

## Data availability

Source data for plots, raw data for counts and intensity measurements, and uncropped gel images generated in this study are provided in a Source data file. The structure factors and coordinates for the KLF4 DBD:KLFA structure have been deposited in the Protein Data Bank under the accession number 6vtx. Source data are provided with this paper.

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

## Acknowledgements

We thank the Baylor College of Medicine Integrated Microscopy Core for the use of the confocal microscopes. We thank Phoebe S. Tsoi for help with DBD protein expression and for reading the manuscript. This work was supported by NIH grants R01 GM122763 to J.C.F. and R21 NS107792 to A.C.F.M. Additional funding was provided by R01 NS105874 and R21 NS109678 to A.C.M.F., by a Core Facility Support Award from the Cancer Prevention and Research Institute of Texas (grant RP160805) to Dr. Martin Matzuk, and by R01 DK121970 and R61 HD099995 to Dr. Feng Li. The ALS-ENABLE beamlines are supported in part by the National Institutes of Health, National Institute of General Medical Sciences, grant P30 GM124169-01 and the Howard Hughes Medical Institute. The Advanced Light Source is a Department of Energy Office of Science User Facility under Contract No. DE-AC02-05CH11231. The Human Stem Cell Core at Baylor College of Medicine is supported in part by the College and NIH grants (P30 CA125123 Osborne and S10 OD028591 Kim).

## Author contributions

A.C.M.F., C.K., K.R.M., and J.C.F. conceived the project. J.J.K., A.C.M.F., C.K., and J.C.F. supervised the experiments. K.J.C. and J.C.F. performed fluorescence cell-based experiments. R.S., M.D.Q., J.C.F., and A.C.M.F. performed in vitro condensation assays; J.C.F. performed the image processing and statistical analysis. H.P., A.L., and J.J.K. performed cell reprogramming experiments. K.J.C. cloned the bacterial and mammalian constructs. R.S., S.S., and J.C.F purified the recombinant proteins. R.S. and C.K. grew crystals and determined the structure; B.S. acquired X-ray data; K.R.M. and C.K. curated the structure. K.R.M. and C.K. developed 3D models for DBD:DNA condensation. M.D.Q. and A.C.M.F. designed, implemented and analyzed single molecule fluorescence experiments. K.R.M. and J.C.F. wrote the manuscript. All authors edited the manuscript.

## Competing interests

The authors declare no competing interests.
