## [Peer Review File · Nature Communications]

Reviewers' comments:

Reviewer #1 (Remarks to the Author):

Authors report on a novel and unexpected mechanism for liquid-liquid phase separation (LLPS) driven by the reprogramming factor KLF4. LLPS has recently been found to represent the dominant mechanism for the formation of nuclear condensates including transcriptional enhancers (in particular super-enhancers). The key components driving LLPS are intrinsically disordered regions (IDRs) that lead to the assembly of heterotypic condensates including TFs and co-activators such as MED and p300. Sharma et al report that in the case of KLF4 the structured DBD alone (lacking IDRs) can mediate LLPS in a DNA dependent manner. DNA methylation further promotes this process. They propose a mechanism whereby a dislodged member of the 3 ZnF array (presumably ZnF1) triggers LLPS. Surprisingly, and contrary to previous reports, Sox2 and Oct4 were found not to be able to initiate LLPS by themselves but need KLF4 to do so. The discovery that IDRs are not essential for LLPS but DNA binding domains too can trigger its formation is a potentially intriguing finding. It may critically impact our understanding of the large C2H2 TF family. I find the subject and claims made in this study exciting and of broad interest. I have however some concerns as to the experimental context, lack of support for key claims and inconsistency with work by others that needs further scrutiny. More evidence should be provided to back up the claims.

Major comments:

1. SOX2 and OCT4 were founded to drive IDR-IDR mediated phase separation in ESCs and in vitro (Boija .. Young et al). Why can they not mediate LLPS in HEK293? I feel the study would benefit substantially if data in a more realistic context such as ESCs could be presented. Likewise, Figure 2g indicates SOX2 and OCT4 proteins do not engage in LLPS in vitro whilst Boija et al report they do. If this is an issue of different experimental conditions (i.e. concentrations, crowding agents), authors should provide details and perform additional control experiments to compare their data with previous reports to resolve discrepancies. For the in vitro experiments this should be straight forward as all required reagents are available.

2. I would like to see more data to back up the claim that ZnF1 can mediate LLPS. Authors suggest that ZnF2 and Znf3 account for the majority of specificity in DNA binding. Do protein constructs lacking ZnF1 still bind DNA but fail to mediate LLPS? If such a construct fails or cannot be expressed how about a construct where ZnF1 residues 6, 3, -1 involved in direct base readout were mutated so they no longer bind DNA but can still promote LLPS ? Such data would further test and could strengthen the model as to a ZnF1 driven LLPS. Couldn't the pose suggested in Figure 3b be a rare exception requiring an atypical configuration of KLFA/KLFB? I would think in most contexts all three fingers are tightly bound to DNA.

3. Can authors rule out that fluorescent proteins affect LLPS? Why was KLF4 tagged with Turq but SOX2/OCT4 with mCherry?

Minor comments:

1. Authors report a novel crystal structure but findings from the structural analysis are not very well integrated into the main text - they are buried in an a very long figure legend in the ED. How different/similar is it to crystal structures previously reported of KLF4 (2WBU, 4M9E,2KE9). Please compare and illustrate DNA contacts made by ZnF1-ZnF3 for these structures in a schematic. The information might be in the long figure legend to ED Fig 14 but a more concise presentation would be

appreciated. Are there any notable differences? Is there any evidence for a detachment of Znf1 in the context of NKA?

2. The ThermOS algorithm predicts – contrary to other motif finders – that the positions bound by ZnF1 are not constrained and can vary: <https://academic.oup.com/nar/article/41/11/5555/2411200>. This notion could be in support of a 'dangling' ZnF1 driven LLPS. Standard databases list a longer motif than the GG[C/T]G core discussed by the authors: <http://jaspar.genereg.net/matrix/MA0039.2/>. It would help to introduce full KLF4 motif and where the ZNFs bind (i.e. in a revised Figure 3).

3. Fig 1b: please clarify in main text how intracellular TF concentration can be reliably controlled.

4. EMSAs ED Fig 5: how often were EMSAs replicated and which binding model was used to estimate the K_d? What is the double band in the 0 μM KLF4-DBD lane for the NKA sample? If this is ssDNA and dsDNA I would be concerned that the ssDNA is bound at nanomolar affinity as well. I can see dimeric and trimeric binding to NKA but not to NKB and NKC. Bound states are seen at 50nM for NKA but at 200nM for other elements. This makes me wonder how reliable the K_d estimates are.

5. Title: I don't think it's surprising that KLF4 can mediate LLPS but if ZNF1 can indeed do this without IDR this could be highlighted more strongly.

Reviewer #2 (Remarks to the Author):

In this manuscript Sharma et al. investigate liquid-liquid phase separation (LLPS) of the transcription factor KLF4. Previously, a number of transcription factors have been shown to be able to undergo LLPS and form condensates or hubs in the nucleus of cells. These condensates/clusters are believed to be important for the regulation of transcription. As such the hypothesis that KLF4, which is well known to bind DNA, is able to undergo phase separation in particular with DNA is reasonable. However, the manuscript lacks the scientific rigor for both the cellular and in vitro work, to be a strong candidate for publication. Instead the authors engage in wide speculations what their observations could mean without the necessary experimental support.

Cell studies/Cell-derived KLF4 (Fig. 1):

Upon overexpression in HEK cells the authors observe some sort of fluorescent foci. However, it remains completely unclear what these foci are and if they have anything to do with liquid-like droplets (also this is claimed). In fact, a number of differently shaped fluorescent structures are observed. The only data which would suggest that these structures are liquid-like is the FRAP data in Fig. 1d. However, a large area with inhomogeneous fluorescence and definitely not representing droplets is bleached there, definitely not supporting that liquid-like droplets have been formed in the nucleus. Additional FRAP data from the different areas (small, large droplets, clusters) are required to demonstrate the formation of droplets. In addition, experimental evidence is required that these fluorescent structures fuse, which is a characteristic of LLPS. Even more important, more work is required what these structures could be and if they are related to any other biomolecular condensates that have previously been observed in the nucleus (i.e. maybe KLF4 just goes into condensates formed by other proteins/nucleic acids).

The fluorescent microscopy data for the purified KLF4-mTurq (Fig. 1g) does not look like droplets, but more like some sort of aggregate/precipitate. Again no data (FRAP, fusion, hexanediol sensitivity, ...) are presented that would demonstrate that this is something related to LLPS.

In vitro phase separation experiments (Fig. 2):

The interpretations almost exclusively rely on fluorescent microscopy images with most of them just using DNA labeling and with basically zero statistics and error estimates. Experiments of multiple independent samples with fluorescently labeled proteins (and mutants), phase contrast microscopy, FRAP, fusion and other experiments are required to characterize the LLPS properties of KLF4 in vitro.

Modeling multivalency (Fig. 3):

Here is where complete fiction starts. The authors just state that multivalency is known to be important for LLPS and then engage in wide speculations how multivalency could be important for phase separation of KLF4. However, there is not a single experimental data set that would show the contribution of multivalency for LLPS of KLF4. Solid experimental data are required here.

What the presented data could mean (Fig. 4):

Again complete fiction not based on the data shown in the manuscript.

Summary.

Our central finding, that the KLF4 tandem zinc finger domain undergoes phase separation with certain DNAs that is strongly enhanced by CpG methylation, uncovers a new type of molecular interaction in the nucleus.

Condensation of tandem ZnF domains with DNA to form droplets requires neither crowding agents nor the KLF4 intrinsically disordered region; it occurs at concentrations as low as 250 nM protein (3 $\mu\text{g/ml}$) and as low as a double digit nanomolar DNA (0.3 $\mu\text{g/ml}$) for short duplexes.

Data from other groups implicate KLF4 in the formation of long-range chromatin contacts. We propose that **KLF4:DNA condensates may organize chromatin by co-localizing genomic sites that recruit KLF4.**

Figures in the current submission that contain **new data** (not merely new presentations) not found in either the main or supplemental Figures of the original submission include:

1. **Fig 1B** (bottom), KLF4-mTurq expression in BJ fibroblasts
2. **Fig 1C** top (and bottom), FRAP of KLF4-mTurq droplets (and clusters) in BJ fibroblasts
3. **Fig 1D**, Fusion of KLF4-mTurq droplets in BJ fibroblasts
4. **Fig 2B**, Distribution of KLF4-mTurq deletion mutants in HEK 293T cells
5. **Fig 2C**, Quantitative analysis of wild type and deletion mutant KLF4-mTurq distribution
6. **Fig 2G**, FRAP of DBD:*NANK* droplets *in vitro*
7. **Fig 2H** (right panel), phase diagram for DBD:*NANK* that extends to 16 nM *NANK*
8. **Fig 3B**, contacts between DBD and *NKA* from our crystal structure
9. **Fig 4F**, single molecule FRET of DBD-induced association between labeled DNAs
10. **Fig 5C** (middle panels), phase diagram for DBD:*NANKm* that extends to 16 nM *NANKm*
11. **Fig 5D**, phase diagrams for mutant DBD with three DNA substrates
12. **Fig 5E**, HEK 293T cells expressing wild type, single mutant, or double mutant KLF4 constructs
13. **Fig 5F**, Quantitative analysis of wild type and point mutant KLF4 distributions in HEK 293T cells
14. **Fig 6G**, Phase separation of 250 nM DBD with long but not short DNAs at 0.6 ng/ μl

New approaches and findings in the current manuscript include that we analyze our fluorescence microscopy data quantitatively, perform a domain analysis in cells that justifies an *in vitro* focus on the DBD, identify point mutations that weaken DBD-mediated phase separation *in vitro*, and show that these mutations decrease puncta formation by full-length KLF4 in cells. Combined with the initially reported result that the enriched phase *in vitro* and the KLF4-mTurq condensate in cells can recruit full-length SOX2 (or OCT4), the biochemical and cellular data drive the narrative in a way that we think makes the work more accessible to readers.

The current manuscript uses additional new data to provide more insights into the basis for DBD:DNA phase separation and help address possible mechanisms. We explore lower concentration regimes and find that CpG methylation decreases the threshold for phase separation even farther than we had previously shown. To examine ideas about multivalency that are important in phase separation by polynucleosomes and by participants in transcriptional activation, we explored longer DNA substrates and showed that phase separation occurs with sub-micromolar DBD and nanomolar or even subnanomolar long DNAs. Our model of DNA bridging by ZnF1 sterically excluded from DBD:*NANK* complexes is supported by our mutational data and by single molecule FRET data showing that DBD induces *NANK* to co-localize in dilute solution, below the LLPS threshold concentrations.

Responses to reviewers.

We present a point-by-point response to the comments of the two reviewers that we received on Aug 29, 2020. Our responses seek to explain how the CURRENT manuscript responds to the original criticisms; we do not re-litigate aspects of the previous manuscript. We usually refer here to Figures in the current submission; occasionally we refer to 'previous figures' from the first submission.

Reviewer 1 was largely receptive to our findings and arguments but frequently asked for more evidence to support our claims and was concerned about inconsistencies with prior reports.

Major comment #1: "SOX2 and OCT4 were found to drive IDR-IDR mediated phase separation in ESCs and in vitro (Boija .. Young et al). Why can they not mediate LLPS in HEK293? I feel the study would benefit substantially if data in a more realistic context such as ESCs could be presented. Likewise, Figure 2g indicates SOX2 and OCT4 proteins do not engage in LLPS in vitro whilst Boija et al report they do. If this is an issue of different experimental conditions (i.e. concentrations, crowding agents), authors should provide details and perform additional control experiments to compare their data with previous reports to resolve discrepancies. For the in vitro experiments this should be straight forward as all required reagents are available."

Re: "SOX2 and OCT4 were found to drive phase separation in ESCs" and "[in this manuscript] SOX2 and OCT4 proteins do not engage in LLPS in vitro whilst Boija et al report they do". Boija et al (2018) show that Med1 and OCT4 co-localize in ESCs, and that OCT4 enhances the formation of small Med1 puncta (especially at super-enhancers). Their data do NOT show that OCT4 independently forms liquid condensates *in vivo*. Boija et al (2018) present *in vitro* experiments performed in the presence of 10% PEG-8000 from which they infer the role of IDR:IDR interactions, but in Fig. 3 they show that **OCT4-GFP does not form droplets on its own in PEG**. However, 10 μ M OCT4-GFP can be **recruited to** 10 μ M Med1-IDR droplets that are induced to form by 10% PEG. Thus, the results of Boija et al parallel what we show: that TF partners such as OCT4 (and, for us, SOX2) can join a pre-existing liquid phase *in vitro*, or be recruited to condensates in cells. Our *in vitro* work tests partitioning of 50 nM labeled OCT4 or 70 nM labeled SOX2 into a phase formed by 9 μ M DBD and 1.5 μ M DNA.

Re: "data in a more realistic context". We seek to explain **early events in reprogramming**, when KLF4 is highly expressed in what are initially differentiated cells; we do not seek to explain the pluripotent state itself, in which KLF4 levels drop compared to early stages of reprogramming. In the current manuscript, we have added expression and co-expression experiments in **BJ fibroblasts**, which are commonly used in reprogramming and so are an appropriate choice. Similar phenomena from both HEK 293T cells and BJ fibroblasts shows that our observations are not dependent on cell type.

Major comment #2: "I would like to see more data to back up the claim that ZnF1 can mediate LLPS. Authors suggest that ZnF2 and ZnF3 account for the majority of specificity in DNA binding. Do protein constructs lacking ZnF1 still bind DNA but fail to mediate LLPS? If such a construct fails or cannot be expressed how about a construct where ZnF1 residues 6, 3, -1 involved in direct base readout were mutated so they no longer bind DNA but can still promote LLPS? Such data would further test and could strengthen the model as to a ZnF1 driven LLPS. Couldn't the pose suggested in Figure 3b be a rare exception requiring an atypical configuration of KLF4/KLFB? I would think in most contexts all three fingers are tightly bound to DNA."

Re: "Authors suggest that ZnF2 and ZnF3 account for the majority of specificity in DNA binding". Previously, we could only 'suggest' this based on prior structural analyses and our own crystal structure. After developing our proposed mechanism, we now **test** this idea: we present new evidence that point mutations to ZnF2 and ZnF3 attenuate both *in vitro* DBD-mediated phase separation (LLPS) and biomolecular condensation of full-length KLF4-mTurq in cells (Fig. 5D-F). We believe that these data now justify our conclusion that cognate KLF4:DNA interactions by the ZnF2 and ZnF3 domains are involved in phase separation of DBD with *NANK* and in KLF4 condensate formation in cells. These data help to rule out the formal possibility that ZnF2 and ZnF3 adopt some unknown, alternate conformation to mediate LLPS with *NANK*.

Re: “Couldn’t the pose in Fig. 3b be a rare exception requiring an atypical configuration of KLFA/KLFB?” We agree that three ZnFs in a tandem array usually contact adjacent regions of the major groove. We think that the special arrangement of KLFA/KLFB overlap forces ZnF1 to be exposed when 3 DBD bind *NANK*, which is more likely when it has been CpG methylated, and that this accounts for the low concentrations at which this 30 bp DNA drives condensation. Our crystal structure shows that ZnF1 makes only one base-specific contact with KLFA (because of the lack of a consensus G base; see response to minor criticism #1), and our structure-based modeling identifies steric clashes that should force ZnF1 to be excluded from such a complex. We now show that overlapping, CpG-methylatable KLF4 sites are also present in the mouse *Nanog* promoter (new Fig. S5), and we argue that such atypical arrangements are conserved because they serve as nucleation sites for KLF4 condensation in a way that is regulated by their CpG methylation state.

We **test** the importance of this ‘atypical configuration’ in three ways:

1. If ZnF1 **were** to be excluded from such a complex, such that it **could** mediate bridging to other duplexes, then the availability of ZnF1 should influence the behavior of the complex in solution, so we use single molecule FRET to test for DBD-mediated interactions between *NANK* molecules. Our new results show (Fig. 4F) that DBD induces association between *NANK* duplexes at concentrations well below the LLPS threshold, demonstrating in dilute solution the type of weak interaction needed to drive LLPS.
2. If exposure of ZnF1 is important to LLPS of DBD with *NANK*, then ratios of DBD:*NANK* that do NOT force exclusion of ZnF1 should not support LLPS. At 2:1 DBD:*NANK*, DBDs should be able to bind at KLFA/KLFC or KLFB/KLFC without incurring clashes, so that all three ZnFs can lie canonically in the major groove (see model in Fig. 4B). As we show in Fig. 5B (top), adding *NANK* to existing DBD:*NANK* droplets actually dissolves them, as we predict from the expected redistribution of DBD into different complexes (and despite the higher total concentration of DNA, which would otherwise be expected to enhance phase separation). Note: the total concentration of ZnF1 has not changed – but its availability has. In Fig. S3, we provide extensive data showing that even at high concentrations, DBD:*NANK* mixtures at ratios lower than 3:1 do NOT lead to phase separation. (This is similar to the behavior of SH3₄ with PRM₄ reported by Rosen and colleagues in (Li 2012)).
3. If the base-specific DBD contacts with DNA that give rise to the 3:1 DBD:*NANK* complex are important to LLPS, and this complex represents an on-pathway species, then DBD mutations that weaken cognate interactions with targets should decrease LLPS. With new data (Fig. 5D vs 5C), we show that point mutations in ZnF2 and ZnF3 that weaken DBD-mediated binding to *NANK* also decrease LLPS with *NANK*.

Major comment #3: “Can authors rule out that fluorescent proteins affect LLPS? Why was KLF4 tagged with Turq but SOX2/OCT4 with mCherry?”

We cannot rule out that the fluorescent tags may contribute in some way; we acknowledge this directly in the new manuscript, and we use it to motivate *in vitro* analysis of tag-free DBD. However, with our new data in Fig. 5E and 5F, we show that single point mutations in KLF4 influence condensation in cells, so apparently any contribution that the tags may make does not overwhelm the role of KLF4 sequence.

We chose mTurquoise2 as a C-terminal fusion tag because it would minimize possible tag homodimerization and spectral overlap with mCherry so that our co-localization experiments using two-channel fluorescence microscopy would have minimal bleed-through. In this manuscript, we now describe repeating the expression experiments with KLF4-mCherry, as indicated in the text, and the results are similar (though KLF4-mCherry does not express as highly).

Minor criticism #1: “Authors report a novel crystal structure but findings... are not very well integrated into the main text - they are buried in a very long figure legend in the ED. How different/similar is it to crystal structures previously reported of KLF4 (2WBU, 4M9E, 2KE9). Please compare and illustrate DNA contacts made by ZnF1-ZnF3 for these structures in a schematic... Are there any notable differences? Is there any evidence for a detachment of ZnF1 in the context of *NKA*?”

We now present the crystal structure in **Fig. 3**, with the requested protein:DNA contact schematic and accompanying text in the main manuscript. One significant contact with DNA differs from previously reported structures. In other structures, ZnF1 K443 (in mouse, K413) contacts a consensus G base in the 8th position of the 9 base JASPAR consensus (see response to minor criticism #3); in *NKA*, this base is a C rather than a G. K443 is disordered in our structure, and no residue makes base-specific contacts with this C; the only base-specific contact made by ZnF1 in our structure is by H446 to the 7th base in the consensus. These data suggest that ZnF1 may interact more weakly with *NKA* than with targets that better match the consensus, but we have no direct evidence for ‘detachment’ of ZnF1 in the context of *NKA*. We thank the reviewer for their valuable suggestion.

Minor criticism #2: “The ThermOS algorithm predicts – contrary to other motif finders – that positions bound by ZnF1 are not constrained and can vary: <https://academic.oup.com/nar/article/41/11/5555/2411200>. This notion could be in support of a ‘dangling’ ZnF1 driven LLPS. Standard databases list a longer motif than the GG[C/T]G core discussed by the authors: <http://jaspar.genereg.net/matrix/MA0039.2/>. It would help to introduce full KLF4 motif and where the ZNFs bind (i.e. in a revised Figure 3).”

Re: “positions bound by ZnF1 are not constrained”. We thank the reviewer for offering an additional line of evidence to support our argument, and we agree that the residues at key positions in ZnF1 do not strongly constrain its ‘choice’ of cognate DNA. Rather than build from this point, however, we prefer to use our structure (see response to minor criticism #1, Fig. 3, and the associated description in manuscript) to argue that ZnF1 may be susceptible to transiently ‘dangling’ from *NKA* because it makes only one base-specific contact, since KLFA has a C rather than the consensus G at the 8th position (base C11 in our structure). With consensus targets, the G at this position is recognized by K443 (though ZnF motif finders indicate that R would better discriminate between C and G). It is tempting to think that the generic nature of ZnF1 might give it minimal bias for the type of interaction that would occur in bridging, but we have no evidence that such interactions would occur in the expected way in the minor groove, so we do not pursue this idea now.

Re: “It would help to introduce full KLF4 motif”. The reviewer is absolutely right. We continue to refer to the GG(T/C)G motif because it is conserved across the three sites in *NANK* and because structural and biochemical data suggest that these bases are most critical to specific recognition, but we should have compared *NANK* to consensus sites. We now present the 9 bp JASPAR consensus motif from ChIP-seq data in Fig. 4, along with a schematic for the arrangement of the KLF4 ZnFs and their contacts with these bases (which can be inferred from prior structures but is also seen in our structure with *NKA*). We use this schematic to suggest that the KLFA and KLFB sites must overlap, which motivates the 3D modeling of DBD at KLF4 sites in *NANK* that identifies large ZnF1:ZnF1 steric clashes. We believe that this substantially improves both the way in which we present our thinking and the actual substance of the argument, and we thank the reviewer for the valuable suggestion.

Minor criticism #3: “Please clarify in main text how intracellular TF concentration can be reliably controlled.”

Our lentiviral approach to co-expression does not allow us to **control** protein levels, but we can quantify expression from the fluorescence of the C-terminal expression tags. Lentivirus does not transduce every cell, and not every transduced cell expresses the same level of protein. The concern of the reviewer is appropriate: although we described the cell-to-cell variation in expression levels in the previous figure 1b, this presentation did not allow a reader (or us, for that matter) to properly assess the significance of that variation.

Our fluorescence microscopy images of transduced cells are subjected to automated image analysis to classify each cell as punctate or not, and we present data for cells with detectable fluorescence in pile-up diagrams according to average fluorescence (see Fig. 1C; also Fig. 5F). The y-axis reveals the variation in total expression; separating the cells into punctate/diffuse piles shows that above some expression threshold cells begin to exhibit the punctate phenotype. The significance of differences in these distributions are assessed using a Student’s paired t-test. We believe that this presentation and analysis strengthens the conclusions that we draw from our in cell deletion analysis and mutational analysis data tremendously.

Minor criticism #4: “EMSA ED Fig 5: how often were EMSAs replicated and which binding model what used to estimate the Kd? What is the double band in the 0 uM KLF4-DBD lane for the NKA sample? If this is ssDNA and dsDNA I would be concerned that the ssDNA is bound at nanomolar affinity as well. I can see dimeric and trimeric binding to NKA but not to NKB and NKC. Bound states are seen at 50 nM for NKA but at 200 nM for other elements. This makes me wonder how reliable the Kd estimates are.”

EMSAs were performed with 2-3 independent replicates. We had previously used a one-site binding model to extract quantitative binding affinities, but in this new manuscript we eliminate the quantitative analysis and merely show representative gels. In the current manuscript, these EMSAs serve to demonstrate that KLF4 is a valid KLF4 target despite matching the 9 base consensus at only 6 positions.

Minor criticism #5: “Title: I don’t think it’s surprising that KLF4 can mediate LLPS but if ZnF1 can indeed do this without IDR this could be highlighted more strongly.”

Our new title focuses on the potential biological significance of the observed phenomena.

Reviewer 2 was receptive to some of the possibilities we sought to examine but much more skeptical of our claims. The reviewer found fault with some of our experimental data, and with the lack thereof, and indicated that models we put forward were not justified from our data, summarizing that in their view:

“...the hypothesis that KLF4, which is well known to bind DNA, is able to undergo phase separation in particular with DNA is reasonable. However, the manuscript lacks the scientific rigor, for both the cellular and in vitro work, to be a strong candidate for publication. Instead the authors engage in wide speculations what their observations could mean without the necessary experimental support.”

The current manuscript contains many new pieces of data and is organized to better explain how our models are based on a combination of both our findings and previous reports from others.

Criticism of figure 1: “Upon overexpression in HEK cells the authors observe some sort of fluorescent foci. However, it remains completely unclear what these foci are and if they have anything to do with liquid-like droplets (also this is claimed). In fact, a number of differently shaped fluorescent structures are observed. The only data which would suggest that these structures are liquid-like is the FRAP data in Fig. 1d. However, a large area with inhomogeneous fluorescence and definitely not representing droplets is bleached there, definitely not supporting that liquid-like droplets have been formed in the nucleus. Additional FRAP data from the different areas (small, large droplets, clusters) are required to demonstrate the formation of droplets. In addition, experimental evidence is required that these fluorescent structures fuse, which is a characteristic of LLPS. Even more important, more work is required what these structures could be and if they are related to any other biomolecular condensates that have previously been observed in the nucleus (i.e. maybe KLF4 just goes into condensates formed by other proteins/nucleic acids). The fluorescent microscopy data for the purified KLF4-mTurq (Fig. 1g) does not look like droplets, but more like some sort of aggregate/precipitate. Again no data (FRAP, fusion, hexanediol sensitivity, ...) are presented that would demonstrate that this is something related to LLPS.”

Re: “The only data which would suggest that these structures are liquid-like is the FRAP data in Fig. 1d.” and “experimental evidence is required that these fluorescent structures fuse”. The current manuscript presents images from two cell types, FRAP data for large round droplets and small clusters, fusion of droplets, and the effects of 1,6-hexanediol on the condensates seen in cells in the panels of Fig. 1. New data not present in the previous version are Fig 1B (bottom), Fig 1C (top and bottom), and Fig 1D. The previous figure 1d (effect of 1,6-hexanediol) is now Fig 1E. Evidence for fusion of KLF4-mTurq-containing droplets in living cells is a timelapse sequence in BJ fibroblasts that is depicted in two dimensions in Figure 1D but validated using 3D z-stacks.

These data establish that the KLF4-mTurq condensates are liquid-like. We do not attempt to infer the presence of LLPS from phenomena observed in living cells, though we think that the 'liquid-like' property we infer from these approaches is important. We agree with the reviewer that we do not know exactly what these foci are.

Re: “more work is required what these structures could be and if they are related to any other biomolecular condensates that have previously been observed in the nucleus (i.e. maybe KLF4 just goes into condensates formed by other proteins/nucleic acids)” The reviewer is correct that KLF4-mTurq may join condensates formed by other species within the nucleus; we cannot exhaustively prove the negative of this possibility. However, the new data we present from cells for the dependence of puncta formation on KLF4-mTurq expression levels and the effects of domain deletions (Fig. 1C), as well as changes to the protein concentration threshold by **single point mutations** that alter DNA recognition (Fig. 5F) indicate that the identity of the KLF4 construct and its total concentration combine to determine if these puncta appear in cells.

Re: “The fluorescent microscopy data for the purified KLF4-mTurq (Fig. 1g) does not look like droplets, but more like some sort of aggregate/precipitate. Again no data (FRAP, fusion, hexanediol sensitivity, ...) are presented that would demonstrate that this is something related to LLPS.” Our new domain deletion analysis in cells (Fig. 1C) provides a rationale for focusing our attention on the DBD rather than the IDR and obviates the need for us to draw any inferences from the behavior of purified full-length KLF4-mTurq, which we have recently established co-purifies with DNA contaminants. Additional new data we present in this manuscript show that very low concentrations of long DNAs drive LLPS with DBD, so we have removed the *in vitro* data for full-length KLF4 from the manuscript.

Criticism of figure 2: “The interpretations almost exclusively rely on fluorescent microscopy images with most of them just using DNA labeling and with basically zero statistics and error estimates. Experiments of multiple independent samples with fluorescently labeled proteins (and mutants), phase contrast microscopy, FRAP, fusion and other experiments are required to characterize the LLPS properties of KLF4 *in vitro*.”

The current manuscript presents fluorescence microscopy *in vitro* co-localization data for labeled DBD and DNA (Fig. 2E), images from a DBD:DNA condensation time course (Fig. 2F), FRAP of tiny droplets *in vitro* with covalent labels for both DBD and DNA (Fig. 2G), and phase diagrams in which the concentrations of DBD and DNA are independently varied and assessed in triplicate, revealing the expected dependence of the lower concentration threshold on protein and DNA levels (Fig. 2H; Fig. 5C,D). Mixtures are classified as condensates or not based on automated analysis of fluorescence microscopy images using the coefficient of variation (CV=std dev/mean) of fluorescence intensity as a measure of inhomogeneity.

The 4-8 fold decrease of the DNA concentration thresholds upon CpG methylation of a cognate KLF4 site in *NANK* (*NANK_m*; Fig. 5C) shows that condensation is sensitive to the molecular identity of the DNA substrate. The increase in protein concentration needed to support condensation of DBD carrying **point mutations** with DNAs (Fig. 5D vs 5C) demonstrates that condensation is very sensitive to the molecular identity of the protein participant. The rapid and complete dissolution of DBD:*NANK* droplets upon adding *NANK* to levels that drop the DBD:DNA stoichiometric ratio from 3:1 to 2:1 (Fig. 5B) establishes that the droplets **equilibrate rapidly** with the solution phase. We believe that these data together firmly establish that DBD:*NANK* undergoes LLPS *in vitro*.

Criticism of figure 3: “Modeling multivalency (Fig. 3): Here is where complete fiction starts. The authors just state that multivalency is known to be important for LLPS and then engage in wide speculations how multivalency could be important for phase separation of KLF4. However, there is not a single experimental data set that would show the contribution of multivalency for LLPS of KLF4. Solid experimental data are required here.”

Having established that DBD undergoes liquid-liquid phase separation with *NANK*, we believe that we should attempt a mechanistic explanation for how this might occur. The literature, including two reviews in 2017 by Brangwynne and by Rosen, indicates that LLPS derives from (i) weak intermolecular interactions that (ii) may be enhanced by multivalency. For many systems, the intermolecular interactions derive from intrinsically

disordered regions, but not always, and we cannot invoke the participation of the 430 residue KLF4 IDR to explain our *in vitro* data because it has been deleted. We therefore seek ways that DBD and DNA might interact other than the simple 1:1 complexes known to form between DBD and a consensus DNA sequence.

The model in the previous figure 3 was intended to argue for a **possible** mechanism by which a 3:1 DBD:*NANK* complex might be able to interact with another DNA duplex; without a physical basis for such interactions, it is not clear how LLPS might occur. The consensus motif, highlighted sequences, and schematic alignment of *NANK* sequence with the KLF4 ZnFs now presented in Fig. 4A summarize our reasons to suspect that the KLFA and KLFB sites of *NANK* are overlapped (these new presentation items were not included in the previous figure 3). The models we present in Fig. 4B-D use superposition of DBD from our crystal structure onto modeled B-form *NANK* DNA to show that the canonical binding mode would cause ZnF1:ZnF1 clashes, as simple linear site overlap in Fig. 4A suggests. Since Fig. 3A shows by EMSA that the overlapped sites are both occupied, we **infer** that the complex includes at least one ZnF1 extending into solution; we **model** this with a previously published crystal structure of DBD bound to a heptamer that does not contact ZnF1. We **propose** that this excluded ZnF1 could mediate contacts with another duplex. To test the ability of the 3:1 complex to interact with another DNA in dilute solution, we labeled the 5' end of the *NANK* coding strand with donor and acceptor fluorophores, reasoning that if there is an exposed ZnF1 that can recruit another duplex, we should detect FRET (see schematic in Fig. 4E). In new data, we show with single molecule FRET methods that at sub-nanomolar DNA concentrations, DBD brings these DNAs together (Fig. 4F). These interaction data establish that DBD can bring together *NANK* DNAs in dilute solution, so something is providing a multivalent interaction; the result is **consistent with** our idea that an exposed ZnF1 can bridge between DNAs, but does not prove that ZnF1 mediates the interaction. Given the rest of the available data, however, we see this as the most likely explanation. The importance of the overlapped sites in *NANK* is also explained in the response to **Major comment #2**, above.

The current manuscript includes new condensation data for DBD with DNAs of different lengths (Fig. 6G) that are consistent with the predicted effects of multivalency. Fig. 6G shows that 250 nM DBD does not condense with 32 nM *NANK* but that it does with the same weight concentration (0.6 ng/ μ l, 2.5 nM) of 404 bp DNA substrate *NP* that includes the *NANK* sequence (three KLF4 sites) and two additional KLF4 consensus sites. The 404 bp DNA also contains many partial matches to the consensus, and Fig. 5A and 5C show that DBD can condense, at higher protein and DNA concentrations, with short DNAs that do not contain cognate KLF4 sites. We argue that the potency with which *NP* (and the even longer *NPE*) condense with DBD arises from multivalency of a long DNA substrate, which can have many DBD bound at cognate and less-than-cognate sites. Multivalency of weak interactions has been shown experimentally and argued theoretically by others to be a basis for lowering a condensation threshold, as we indicate and cite in the text.

Criticism of figure 4: "What the presented data could mean (Fig. 4): Again complete fiction not based on the data shown in the manuscript."

Others have shown that KLF4 binds to CpG methylated closed chromatin early in cell reprogramming. Published data from ChIPseq and chromosome capture experiments by several groups implicate KLF4 in chromatin structure: at tens of thousands of PSC-specific contacts, at both repressing and activating chromatin loops, and at enhancers and super-enhancers. We propose a model in which **the physical ability of KLF4 to condense with DNA enables it to organize DNA**. We speculate (in the Discussion and Fig. 7) as to how the properties we have shown for DBD:DNA might enable full-length KLF4 to behave as other groups have shown that it does.

The image of our previous model attempted to portray how KLF4:DNA condensation could not only organize DNA but help to recruit the machinery of transcriptional activation. While we still address this in the text because it fits nicely into existing ideas about IDR-mediated control of transcriptional initiation, and importantly does not conflict with them, in Fig. 7 and in the title of the paper we focus on how KLF4 might organize DNA through biomolecular condensation, since we believe that our model fills a lacuna in our understanding of chromatin organization.

REVIEWER COMMENTS

Reviewer #1 (Remarks to the Author):

Sharma et al have submitted a substantially revised manuscript containing new experiments to back up their claim that the DBD of KLF4 is capable to mediate LLPS in vitro as well as in two human cell lines. I appreciate that new data in more relevant fibroblasts cell lines were added – although authors did not make an effort to convert them into iPSCs and monitor phase separation during reprogramming. Newly added single molecule FRET experiments suggest that the three-finger C2H2 domain ‘bridges’ remote DNA elements. This bridging points to a mechanism how an IDR independent TF binding site clustering and phase separation could be brought about. I remain intrigued by the novel and unexpected model that the DBD itself can foster KLF4:DNA condensation and appreciate that this could impact our view on how TFs operate in the context of chromatin and in particular during cell fate switching. However, I remain equally skeptical whether the in my view still rather vague and correlative data source is solid enough to provide a strong backing of such claims. I am leaning towards giving the authors the benefit of the doubt and let the scientific community further evaluate the merit of the work after publication. After all, paradigms such as the one that IDRs are the key for the self-association of TF/DNA complexes whilst the DBD directs them to relevant loci are there to be challenged. Nevertheless, there are a couple of points – some I had suggested in the first round of review – that should be addressed experimentally or through re-analysis of public data. I feel with the assays at hand and more carefully crafted controls (different DNA elements, KLF4 mutants) authors could go a long way to test their model more rigorously.

Major comments:

1. I appreciate the efforts made by the authors to evaluate mutant KLF4 constructs. However, I don't see the reasoning behind the choice of mutants: one ZnF2 mutant construct should affect selective binding to methylated DNA, another mutant in ZNF3 is expected to reduce affinity and then there is a double mutant of both these modification. Authors suggest the floppy Znf1 is key to drive the condensation. Why not engineering this finger by (i) increasing it's affinity for cognate DNA (i.e. Arginine at -1,3,6), increasing floppiness by substitution residues that in the structure still bind DNA and (iii) deleting Znf1 (or severely compromising it). Such constructs could inform the mechanism for the condensation in vitro and in the cell lines. The present constructs merely reduce DNA binding so it's not surprise that LLPS (and bridging) is lost. As the model and claims are centered on Znf1 this module should be at the center of attention.
2. Why have all in vitro experiments been carried out with degenerate KLF4 binding sites? Would authors expect that a cognate KLF4 site abolishes droplet formation as it prevents ZNF1 from ‘dangling’? Along these lines: would authors consider to present genome wide analyses (i.e. be re-analyzing published KLF4 ChIP-seq data) to clarify how common the pose of the Nanog promoter is? If we don't need exactly this pose for LLPS why not adding in data with other clustered KLF4 binding sites and show that they behave similar as the Nanog promoter? I note data with ‘non-cognate sites’ but for these sites, I assume, we have no model to link structural poses to LLPS.
3. I find the ZNF ‘bridging’ model quite interesting. Yet, I would rather think that proteins with a larger array of C2H2 modules are capable of this feat (i.e. CTCF, REST). Can constructs with 1-2 C2H2 domains bind DNA (I surmise they cannot). Do authors believe the ‘bridging’ is driven by protein-protein interactions between juxtaposed ZNF1? I feel if authors could shed light onto the molecular basis for the ‘bridging’ the manuscript would tremendously gain in value.

Minor comments:

-please use the common -1,2,3,6 nomenclature in display items of structural models for easier cross-referencing to the C2H2 recognition code. Include the nomenclatures in the main text too when mutants are described.

-I expect the DBD of KLF4 to bind DNA in low nanomolar affinity. The EMSAs in the supplement indicate a ~500 nM Kd. Please discuss reasons for this discrepancy. High quality reagents are key for the various in vitro assays presented. The fact that authors could crystallize the DBD suggests reagents are of high quality.

-Authors say that fibroblasts lack KLF4. I would argue that these cells (at least in mice) do express reasonable levels of KLF4

-Figure 2B should include images for FL protein and ideally data for the re-engineered (and deleted) ZNF1.

-protein production and purification procedure lack detail and should be expanded

-can it be illustrated more clearly how 'puncta' and 'droplets' differ in microscopic images?

Reviewer #2 (Remarks to the Author):

The authors revised their manuscript in light of the referees' comments. They performed new experiments, additional statistics and greatly improved the manuscript text and interpretation. The revised version of the manuscript fully addresses my concerns. Congratulations to the authors for an excellent work.

We thank Reviewer #2 for their clearly expressed appreciation of the changes relative to the prior submission, and we note that they accept the manuscript without caveat. Reviewer #1 expresses questions and suggestions that challenge us to better present our arguments. We thank them for this close criticism. Reviewer #1 also suggests additional experiments that we believe extend the scope of the work without directly addressing our central conclusions. Though of interest and importance, for us they fall outside the scope of this submission. Our responses follow.

Reviewer #1 (Remarks to the Author):

Sharma et al have submitted a substantially revised manuscript containing new experiments to back up their claim that the DBD of KLF4 is capable to mediate LLPS in vitro as well as in two human cell lines. I appreciate that new data in more relevant fibroblasts cell lines were added – although authors did not make an effort to convert them into iPSCs and monitor phase separation during reprogramming.

We have made efforts to track KLF4 phase separation during reprogramming in collaboration with our co-author Jean Kim, who directs the BCM Human Stem Cell Core. Unfortunately, expression from the transiently transformed KLF4-mTurq expression vectors does not persist through the 21 day reprogramming process; we will need CRISPR-modified cell lines to visualize KLF4 levels in **live** cells in this process. However, we examined expression and distribution of endogenous KLF4 in **fixed** cells using anti-KLF4 antibodies, and in mid- or late-reprogramming KLF4 expression in fibroblast-like cells is low, whereas in cells whose morphology indicate that they are acquiring pluripotency we see high levels of KLF4 (see **Figure A**; this is already known in the literature). We are reluctant to interpret the granular nature of KLF4 staining as phase separation, since the cells are fixed. We do not believe that including this data would strengthen the paper, but we share it here.

Figure A. Additional KLF4 enhances BJ fibroblast reprogramming.

On Day 0, cells are treated with CytoTune-iPS 2.1 Sendai Reprogramming Kit (Invitrogen) vectors for KOSM (KLF4, OCT4, SOX2, c-MYC) only or supplemented with an additional KLF4 vector. **(i)** Bright field microscopy images of BJ fibroblasts treated to generate iPSCs. Some of the elongated fibroblasts adopt spherical morphologies at Day 7, initiating what will be an iPSC colony. The extra KLF4 vector enhances the number and size of early pre-iPSC colonies (right panels vs left). **(ii)** Cells were fixed at different days during reprogramming and stained with anti-KLF4 antibodies (green). Fluorescent image of a zoomed in region of an early stage iPSC colony shows high levels of KLF4 expression in the colony and much lower (but detectable) KLF4 expression in the surrounding elongated BJ fibroblasts, suggesting that high KLF4 expression correlates with early reprogramming success. **(iii)** Alkaline phosphatase staining of treated cells to identify iPSC colonies. Consistently, after 21 days, cells that had been treated with the additional KLF4 vector show higher populations of cells that are more completely reprogrammed (darker stain).

Newly added single molecule FRET experiments suggest that the three-finger C2H2 domain 'bridges' remote DNA elements. This bridging points to a mechanism how an IDR independent TF binding site clustering and phase separation could be brought about. I remain intrigued by the novel and unexpected model that the DBD itself can foster KLF4:DNA condensation and appreciate that this could impact our view on how TFs operate in the context of chromatin and in particular during cell fate switching. However, I remain equally skeptical whether the in my view still rather vague and correlative data source is solid enough to provide a strong backing of such claims.

We are unsure if this qualified praise requires a response, and we are happy that the reviewer sees “*that this could impact our view on how TFs operate in the context of chromatin and in particular during cell fate switching*”. We note that the reviewer seems to conflate the basic experimental findings with models that seek to provide explanations of those findings. That “*the DBD itself can foster KLF4:DNA condensation*” is established by our extensive data *in vitro*, and is supported by additional data in cells; drawing this (surprising) **conclusion** does not require any model. However, merely demonstrating phase separation does not explain **how** it might occur, and the surprising nature of the result challenges us to provide some kind of plausible mechanism for this unprecedented observation, if possible.

Having established **that** this physical phenomenon occurs, we developed a possible model for **how** it might occur ('bridging', which is suggested by the arrangement of KLF4 sites in the *NANOG* promoter) and we devised a way to test for bridging in dilute solution (single molecule FRET experiments). We are not claiming that no other model is possible – we are providing a feasible model for a surprising finding. We have considered other models (as discussed below with regards to possible roles for ZnF1:ZnF1 interactions); no other model is consistent with all the data. And, of course, no other model exists in the literature, since the observations here that require a new model are otherwise unknown.

One importance of the model is that it allows the reader to imagine how other tandem C2H2 ZnF proteins might help organize chromatin; this may explain why the reviewer wishes the model to be as detailed and developed as possible. But our experiments are focused on KLF4, for which the simple model as presented suffices as a plausible explanation.

I am leaning towards giving the authors the benefit of the doubt and let the scientific community further evaluate the merit of the work after publication. After all, paradigms such as the one that IDRs are the key for the self-association of TF/DNA complexes whilst the DBD directs them to relevant loci are there to be challenged. Nevertheless, there are a couple of points – some I had suggested in the first round of review – that should be addressed experimentally or through re-analysis of public data. I feel with the assays at hand and more carefully crafted controls (different DNA elements, KLF4 mutants) authors could go a long way to test their model more rigorously.

We see our findings as supplementary to the IDR-mediated paradigm for transcriptional activation; we do not argue AGAINST that paradigm in the paper, and we include that paradigm in our speculative cartoon in Figure 7. We agree that more experiments could be undertaken to test our model, and more narrowly define the mechanism, but because we have provided a plausible model to explain our remarkable observations, and we have performed a reasonably strong test (smFRET) of one of its predictions (bridging), we believe that there is no reason for the lack of such experiments to hold up the publication of the initially unexpected and still-unknown-to-the-rest-of-the-world result that **the KLF4 DBD mediates condensation with DNA in the absence of the IDR.**

Major comments:

1. I appreciate the efforts made by the authors to evaluate mutant KLF4 constructs. However, I don't see the reasoning behind the choice of mutants: one ZnF2 mutant construct should affect selective binding to methylated DNA, another mutant in ZnF3 is expected to reduce affinity and then there is a double mutant of both these modification... The present constructs merely reduce DNA binding so it's no surprise that LLPS (and bridging) is lost.

We thank the reviewer for pushing us to better explain our thinking on this matter.

The initial KLF4 phase separation with DNA surprised us; when first faced with this result, we sought to challenge it, as we expect it will be challenged by many first-time readers (and reviewers). We first made mutations to determine if the phase separation observed *in vitro* relied in any way on well-known, canonical KLF4:DNA contacts, reasoning that if it did, condensation could not be some artifact that we were somehow inadvertently generating. We focused on residues that show base-specific interactions in our crystal structure; this led us away from ZnF1. Showing that **large-to-small point mutations in ZnF2 and ZnF3 abolish or substantially weaken phase separation** eliminates the possibility that a trace of improperly folded DNA binding domain, or a completely different mode of interaction between the domain and DNA, or a trace of some other unknown macromolecular contaminant, is responsible for the observed condensation. These mutations are **critical** to validating the central observation. Subsequently, these point mutations serve as important **tools** for us to assess the significance of puncta or droplets in cells, where we have no ability to control for other molecular participants. The altered expression threshold for puncta formation in cells by full-length KLF4 carrying one or two point mutations at C2H2 ZnF selectivity positions, which must only subtly change the conformation of the protein as a whole, strongly indicates that **the folded KLF4 DBD makes canonical interactions with DNA through ZnF2 and ZnF3 to induce the observed puncta**. This is **critical** to establishing the biological relevance of the observed condensates in cells, and to advancing the argument that condensate formation in cells could exhibit selectivity.

We have rewritten this part of the text to emphasize the rationale for these mutations and the critical significance of the experimental outcomes. Where we first introduce mutational analysis, we added the sentence, "On the other hand, if the observed condensation depends on an unfolded DBD fraction, a different DBD surface, or a trace contaminant, then large-to-small mutations at the 'specificity residues' of the C2H2 recognition code³⁹ should have no effect on LLPS". We extended the sentence "We infer that the DNA-contacting surfaces of ZnF2 and ZnF3 are therefore important to condensation mediated by full-length KLF4 in cells" with "and that the observed condensation is likely mediated by KLF4 molecules whose DNA binding domain is properly folded."

As the model and claims are centered on ZnF1 this module should be at the center of attention. Authors suggest the floppy ZnF1 is key to drive the condensation. Why not engineering this finger by (i) increasing its affinity for cognate DNA (i.e. Arginine at -1,3,6), increasing floppiness by substitution residues that in the structure still bind DNA and (iii) deleting ZnF1 (or severely compromising it). Such constructs could inform the mechanism for the condensation in vitro and in the cell lines.

These questions are interesting, but our primary claims are not centered on ZnF1, which is not mentioned in the abstract. Our primary claim is that **the KLF4 DBD mediates condensation with DNA in the absence of the IDR**. Our attempt to seek a mechanistic basis for this phenomenon uncovered the idea that ZnF1 should be sterically excluded from *NANK*, which led to the idea of bridging, which we tested. We cannot see how the outcomes of the interesting experiments asked by the reviewer regarding ZnF1 would cause us to alter any of the conclusions drawn in this manuscript – and so although they would almost certainly lead us to new, additional insights, they are beyond the scope of this manuscript.

In response to the request from reviewer #1 that we delete ZnF1, we have truncated the *in vivo* expression construct from DBD-mTurq to just ZnF2-ZnF3-mTurq, and this abolishes puncta formation (see **Figure B**). Because this construct expresses at a very high level its behavior cannot be readily compared with Fig 5E and 5F or Fig 2B in the main manuscript. We stand by the basic conclusion that puncta formation is lost, since expression is always diffuse, but we do not understand the different levels of expression. We do not plan to include these data in the manuscript, since we have not tested the ZnF2-ZnF3 construct for condensation *in vitro*, and since for us the primary significance of condensation by KLF4^{ΔIDR}-mTurq is that the IDR is dispensable.

Figure B. Further truncating KLF4^{ΔIDR}-mTurq abolishes condensation. Fluorescence microscopy of live HEK293T cells shows that a construct containing the KLF4 DNA binding domain fused to mTurq expresses moderately and shows puncta or droplets (left, red arrows), but although the ZnF2-ZnF3-mTurq construct can express at very high levels in cells, at all detectable expression levels the distributions are uniform, with no indication in hundreds of transformed cells of a punctate or droplet phenotype. (Compare to Figure 2B.)

2. *Why have all in vitro experiments been carried out with degenerate KLF4 binding sites? Would authors expect that a cognate KLF4 site abolishes droplet formation as it prevents ZnF1 from 'dangling'? Along these lines: would authors consider to present genome wide analyses (i.e. be re-analyzing published KLF4 ChIP-seq data) to clarify how common the pose of the Nanog promoter is? If we don't need exactly this pose for LLPS why not adding in data with other clustered KLF4 binding sites and show that they behave similar as the Nanog promoter? I note data with 'non-cognate sites' but for these sites, I assume, we have no model to link structural poses to LLPS.*

The 'degenerate binding sites' in our experiments are a stretch of native sequence from the human *NANOG* promoter that contains three KLF4 sites. Some aspect of the arrangement of these sites is important to condensation, because unrelated DNAs of similar length phase separate only at much higher concentrations, as we show. We also show that the selectivity of DBD:DNA interaction is important to condensation, because CpG methylation of the central KLF4 site strongly enhances both DNA binding and phase separation. We acknowledge that we have not dissected this system fully, but **we have established that this novel phenomenon shows strong DNA sequence selectivity.**

If KLF4 DBD could phase separate equally potently with any DNA, then it could only be important to organizing chromatin *in vivo* if some other region of KLF4, or some other partner, were to provide site specificity and selectivity. Our fortuitous discovery of a DNA sequence that strongly potentiates KLF4 LLPS relative to other DNAs shows that **the DBD:DNA interaction itself can provide selectivity**, and that it is sensitive to CpG methylation. That this sequence element is within the *NANOG* promoter, which is critically regulated by KLF4, establishes the importance of this selectivity and elevates our finding from merely a phase phenomenon to a plausible mechanism for organizing chromatin. (Arguing this point here has led us to use it to open the Discussion section, with the sentence, "Our demonstration that the KLF4 DBD condenses much more potently with certain DNA sequences implicates KLF4:DNA interaction in directing condensation of full-length KLF4 at particular regions of the genome.") Given our data that establishes the very low concentration thresholds for condensation of long DNAs, we agree with the reviewer that other sites that potently drive KLF4 condensation exist in the genome, but finding them (or failing to) will not alter the importance of our current findings.

We do expect that other arrangements of KLF4 sites may potentiate LLPS, and so reanalyzing the published KLF4 ChIP-seq data, as requested by reviewer 1, makes sense for the field. Indeed, we

expect that this manuscript will be heavily cited by those who undertake such analyses. Since not every documented KLF4:DNA contact needs to arise from condensation (some may be simple binding events), there is a lot to tease apart. We have initiated such analyses with additional collaborators, and this will represent a large amount of work. We believe that our measurements with a variety of short or long, cognate- or non-cognate, methylated or unmethylated DNA substrates suffice, for this publication, to establish that the DBD:DNA interaction can be very specific and selective.

We agree with the reviewer that we do not have structural models for how DBD induces condensation of non-specific DNA. However, the bridging model can be readily adapted. Perhaps some DNAs are bound more predominantly by ZnF1-ZnF2, and ZnF3 abandons the major groove to contact another duplex and 'bridge'. Because this idea is of interest for the physical model, but not really for the biological importance of KLF4, we have not pursued it in the body of the text. In the discussion, where we raise the family of tandem ZnF proteins, we point out that many of these have been shown to be targeted to their genomic sites using only a subset of their ZnFs, and we propose that the other ZnFs could support bridging. We do not think that there is anything special about ZnF1 that enables it to support bridging; though we may be wrong about that, we are not making a claim one way or another.

3. I find the ZnF 'bridging' model quite interesting. Yet, I would rather think that proteins with a larger array of C2H2 modules are capable of this feat (i.e. CTCF, REST). Can constructs with 1-2 C2H2 domains bind DNA (I surmise they cannot)? Do authors believe the 'bridging' is driven by protein-protein interactions between juxtaposed ZnF1? I feel if authors could shed light onto the molecular basis for the 'bridging' the manuscript would tremendously gain in value.

We agree that other tandem array C2H2 ZnF proteins might also 'bridge', and we expect that others working with some of the 800 or so human versions of these proteins will test this, and cite this work when they do so. Indeed, this potential implication of our findings is one of the reasons it deserves to be communicated prominently. Like the reviewer, we expect that a protein comprising only 1 or 2 ZnFs would not bridge; but if the protein had other domains it might achieve multivalency some other way, for instance by interactions of its IDR, and so be able to 'bridge' as a multimer, and participate in condensation.

The second question about ZnF1:ZnF1 interactions is a good one, but unfortunately we have little experimental information on which to base a response. There is a modest-sized literature on ZnF-mediated protein:protein interactions, and we have spent time sketching out models that would exploit this to drive phase separation. However, we do not think that a typical ordered protein:protein interaction between ZnF1 domains is the best explanation for the observed condensation, which requires multivalency (see refs 50-51 in the current manuscript).

If two 3:1 DBD:DNA complexes were brought together by ZnF1:ZnF1 interactions, they could form a dimer of complexes, but no interaction domain would be available to mediate formation of a higher order assembly: in this model, a pair of complexes would mediate just one DNA:DNA bridging event; if these both require special arrangements of KLF4 sites, they would be very rare events. Weak, non-specific interactions between the exposed ZnF1 of one DBD:DNA complex and any region of the DNA that flanks a second DBD:DNA complex can drive association while leaving the exposed ZnF1 of that second complex free to interact with another DNA strand that may have a third complex, and so on, to promote condensation; each complex would mediate one DNA:DNA bridging event using only one 'special' KLF4 DNA binding site. This second model (our current model) explains the very low DNA concentration thresholds for long DNA substrates better than the first model.

We believe that our size exclusion chromatography data (not shown) rules out any tendency for tight ZnF1:ZnF1 interaction in solution, but it is possible that in a DNA bridging mode, ZnF1 could be presented in a manner or context that would enhance its ability to interact with another ZnF1, or perhaps with some surface of ZnF2-ZnF3:DNA complex that it cannot 'reach' on its own monomer. Since we have no data to indicate this but cannot rule out the possibility at present, we do not raise

the matter in our discussion. We are of course intensely focused on getting crystal structures for bridging species, which would allow us to design mutations (in ZnF1 or elsewhere) to perturb bridging and condensation. There is much to explore about the physical basis for bridging, and we think that the smFRET approaches we developed here to test for bridging will be important in defining the mechanism, but the experiments are beyond the scope of this paper.

Minor comments:

-please use the common -1, 2, 3, 6 nomenclature in display items of structural models for easier cross-referencing to the C2H2 recognition code. Include the nomenclatures in the main text too when mutants are described.

We have made the requested changes to the text and to Figure 3, and we agree that using this nomenclature makes the experimental design and results more accessible to a first-time reader. We thank the reviewer for this helpful suggestion. We have added a citation for the C2H2 recognition code, and we refer to it when we discuss the design rationale for loss-of-DNA-binding mutations.

-I expect the DBD of KLF4 to bind DNA in low nanomolar affinity. The EMSAs in the supplement indicate a ~500 nM K_d . Please discuss reasons for this discrepancy. High quality reagents are key for the various in vitro assays presented. The fact that authors could crystallize the DBD suggests reagents are of high quality.

We assay KLF4 DBD DNA binding using EMSA, which is a non-equilibrium technique, under conditions that facilitate quantification and direct visualization of the number of bound states; buffer, glycerol, or temperature changes can substantially affect the absolute observed apparent K_d s. Reported KLF4 DBD:DNA K_d s are in the double digit nanomolar range and were performed with a fluorescence anisotropy (equilibrium measurements) for short fluorescently labeled DNA duplexes with DBD under conditions similar to ours (Liu et al 2014, ref 17 in the current manuscript). There are many reports in the literature on the discrepancy between K_d measurements using EMSA vs fluorescence anisotropy; since we are not relying on quantitative measures of binding, we choose not to reiterate these points in the paper.

We eliminated quantitative analysis of DNA binding in the current version of the manuscript because the models required to fit binding to NANK (multi-site, cooperative and anti-cooperative) are very complex. We still include the EMSA images to establish that DBD interacts with both cognate and non-cognate DNA sufficiently tightly to cause mobility shifts, and to show the number of shifted species that are generated.

-Authors say that fibroblasts lack KLF4. I would argue that these cells (at least in mice) do express reasonable levels of KLF4

We examined expression and distribution of endogenous KLF4 in fixed cells during a reprogramming time course using anti-KLF4 antibodies; these data show that KLF4 expression in fibroblast-like cells is very low, but that cells whose morphology indicate that they are acquiring pluripotency express high levels of KLF4 (see Figure A; this is already known in the literature).

-Figure 2B should include images for FL protein and ideally data for the re-engineered (and deleted) ZnF1.

The images for full-length KLF4-mTurq are in Fig 1A; we prefer the layout of Fig 2 without replicating the three panels in Fig. 2B.

-protein production and purification procedure lack detail and should be expanded

The text has been expanded to describe the purification and refolding process in more detail.

-can it be illustrated more clearly how 'puncta' and 'droplets' differ in microscopic images?

Figure 1 illustrates the difference; our definitions of 'droplet' and 'puncta' based on morphology are in the supplement, but these could be moved to the body of the paper. Puncta are irregularly shaped, or so small that we cannot determine their shape; droplets are characterized by sphericity. In Supplementary fig. 2 we define small droplets as having circularity >0.8 and diameter $0.3-1 \mu\text{m}$, and large droplets as having diameter $>1 \mu\text{m}$.

REVIEWERS' COMMENTS

Reviewer #1 (Remarks to the Author):

I agree with the most relevant points raised by the others in their rebuttal letter and appreciate the adjustments made in this second round of revision. In view of the novelty and the unexpected nature of the findings I support publication in its current form even if not all suggestions were considered. I trust the authors will work on follow up studies to further test their model that modular DBDs could drive LLPS by bringing remote DNA sequences together and in fact do not require IDRs to achieve this.

Minor comments:

1. Panel 2I is not marked in the Figure
2. I believe the amino acids sequence of the DBDs of mouse and human KLF4 are identical. This should be made clear when the structural data in Figure 3 are discussed as it currently reads as if authors elucidated a novel human structure extending previous mouse structures. What's structurally novel is the NKA sequence element.
3. Please mark KLFA, KLFB, KLFC in the cartoons of Figure 4.